# Zipfian Whitening

**Sho Yokoi**
Tohoku University / RIKEN
yokoi@tohoku.ac.jp

**Han Bao**
Kyoto University
bao@i.kyoto-u.ac.jp

**Hiroto Kurita**
Tohoku University
hiroto.kurita@dc.tohoku.ac.jp

**Hidetoshi Shimodaira**
Kyoto University / RIKEN
shimo@i.kyoto-u.ac.jp

## Abstract

The word embedding space in neural models is skewed, and correcting this can improve task performance. We point out that most approaches for modeling, correcting, and measuring the symmetry of an embedding space implicitly assume that the word frequencies are *uniform*; in reality, word frequencies follow a highly non-uniform distribution, known as *Zipf's law*. Surprisingly, simply performing PCA whitening weighted by the empirical word frequency that follows Zipf's law significantly improves task performance, surpassing established baselines. From a theoretical perspective, both our approach and existing methods can be clearly categorized: word representations are distributed according to an exponential family with either uniform or Zipfian base measures. By adopting the latter approach, we can naturally emphasize informative low-frequency words in terms of their vector norm, which becomes evident from the information-geometric perspective [42], and in terms of the loss functions for imbalanced classification [36]. Additionally, our theory corroborates that popular natural language processing methods, such as skip-gram negative sampling [37], WhiteningBERT [26], and headless language models [23], work well just because their word embeddings encode the empirical word frequency into the underlying probabilistic model.

 https://github.com/cl-tohoku/zipfian-whitening

## 1 Introduction

Representing discrete words by continuous vectors is a fundamental and powerful framework of modern deep-learning-based natural language processing (NLP). Static word embeddings [43, 37], dynamic word embeddings [18, 33], and causal language models [45, 12, 54] have caused a paradigm shift—they have greatly improved the performance of virtually all kinds of NLP applications and have been actively used in relevant areas as well. While the embedded units may be characters or subwords instead of words, we simply refer to them collectively as *word*.

Recently, the machine learning and NLP communities have discovered that the word embedding space is "skewed" and that correcting this can lead to better performance in downstream tasks [39, 21, 16, 56]. The isotropy of the embedding space would be one factor: vectors dispersing more evenly should be more discriminative than those clustered in the same direction [38, 21, 51]. Typically, such spatial symmetry in the embedding space is enhanced through centering/whitening [39, 16, 26].

Nevertheless, we would like to point out that most existing approaches implicitly assume *uniform* word frequency to formalize spatial symmetry. Consider the classical centering operation as an example: we first calculate the mean of the word vectors, and then subtract it to ensure they are zero-meaned. This method, however, has an unexpected pitfall. Recall that the definition of the centroid or barycenter of a random vector $x \sim p$, assuming it has a finite set of distinct realizations,

38th Conference on Neural Information Processing Systems (NeurIPS 2024).

is given by $\mathbb{E}_{\boldsymbol{x}\sim p}[\boldsymbol{x}] = \sum_i p(\boldsymbol{x}_i)\boldsymbol{x}_i$. The classical centering, based on the standard (unweighted) mean, implicitly assumes that all words occur uniformly $p(\boldsymbol{w}_1) = \cdots = p(\boldsymbol{w}_n)$. In reality, however, word frequencies are known to follow a highly non-uniform distribution[1], creating a significant gap between the methodology and the actual usage of words. This seemingly obvious issue does not arise when addressing classical statistical estimation problems, as data vectors in our hands are usually representations of observations or instances. In contrast, word vectors used in NLP are representations of types or classes; each of them (such as the vector for 'the') abstracts the numerous instances (such as the tokens of 'the') appearing in the data. This problem of *hidden* frequencies becomes apparent in the cases where the type-token distinction [58] is crucial, such as when dealing with natural language data (§ 2). The take-home message of this paper can be summarized as follows: use empirical word frequencies when calculating expected values. Following this very simple guideline leads to strong empirical outcomes (§ 3.2, § 3.3) and opens a rich theoretical landscape (§ 4, § 5).

**Notation** Let $\mathcal{V} = \{w_1, \ldots, w_n\}$ denote the vocabulary, i.e., the set of words in interest. Bold-face $\boldsymbol{w}_i \in \mathbb{R}^d$ denotes the row vector of each word type $w_i$, and $p(w_i) \in [0, 1]$ denotes its frequency.

## 2  Motivation: type-token distinction and expected values

Why have word frequencies been overlooked when considering the geometric properties of embedding spaces? This can be explained through the concept of *type-token distinction* [58], which is a fundamental concept in linguistics and related fields but generally not required in statistical machine learning. Here, **type** represents a class and **token** represents an instance. For example, the phrase 'perform natural language processing in a natural way' contains eight tokens and *seven* types. The instances 'natural' appear twice, but as a word type, it is counted only once.

With the type-token distinction in mind, let us take a fresh look at data matrices and their expected values. Typically, each row in a data matrix represents one observation, i.e., one instance **token**. If we want to centralize a set of data vectors, computing the unweighted mean is a natural way in the machine learning pipeline. On the other hand, each row of a word embedding matrix, i.e., word vector, is a **type** embedding. Each word vector abstracts the numerous instances appearing repeatedly in a corpus, though information on the frequency of instances for each word type is not encoded in it. The unweighted mean of word vectors treats type vectors as token vectors, resulting in the complete omission of word frequency information.

Let us describe the above idea formally. The data matrix $\boldsymbol{X} \in \mathbb{R}^{n \times d}$ or the set of data vectors $\{\boldsymbol{x}_i\}_{i=1}^n \subseteq \mathbb{R}^d$ represents a collection of instances, observations, or **tokens**; then the empirical distribution is $\mu_{\boldsymbol{X}} = \sum_{i=1}^n \frac{1}{n}\delta(\boldsymbol{x}_i)$, where $\delta$ is the Dirac delta function. Here, the unweighted mean can be seen as the expectation $\widehat{\mathbb{E}}_{\boldsymbol{x}\sim\mu_{\boldsymbol{X}}}[\boldsymbol{x}] = \sum_{i=1}^n \frac{1}{n}\boldsymbol{x}_i$ with the empirical distribution. On the other hand, the word embedding matrix $\boldsymbol{W} \in \mathbb{R}^{n \times d}$ or the set of word vectors $\{\boldsymbol{w}_i\}_{i=1}^n \subseteq \mathbb{R}^d$ represents a collection of **types**. When describing the empirical distribution, the hidden frequency $p$ of tokens is necessary. Given $p$, the empirical distribution is $\mu_{\boldsymbol{W}} = p(w_i)\delta(\boldsymbol{w}_i)$. From this perspective, the centroid of the word vectors should be written as the expectation $\widehat{\mathbb{E}}_{\boldsymbol{w}\sim\mu_{\boldsymbol{W}}}[\boldsymbol{w}] = \sum_i p(w_i)\boldsymbol{w}_i$ over $p$.

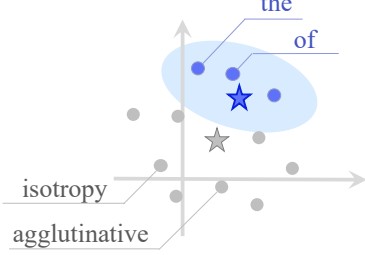

Figure 1: Low-frequent words {●} and high-frequent words {●} are unevenly distributed in the embedding space [39, 24, 44, 10]. Consequently, the "apparent" mean calculated by unweighted averaging ☆ often differs from the actual centroid ★.

The distinction is not just "theoretical." First, refer to Fig. 1. Word vectors are known to cluster by frequency [39, 24, 44, 10]. In this situation, the centroid ★ weighted by the word frequencies is located near the narrow region where high-frequent words are concentrated (a region with a light blue background), and thus differs from the unweighted mean ☆. Second, see Table 1, which shows

---

[1]As known as Zipf's law. If we count the frequencies of words in huge English corpora, we find that 'the' has a frequency of about $5.89 \times 10^{-2}$ and 'isotropy' has a frequency of about $3.47 \times 10^{-8}$, a difference of a million times greater.

50 words sampled from each of types and tokens. Uniform sampling from types, corresponding to an unweighted mean, tends to select mostly rare words from the heavy tail. Sampling from tokens clearly captures a more natural representation of language as it typically appears in text.

Table 1: The difference between type-based sampling and token-based sampling.

| Words sampled from types | Words sampled from tokens |
|---|---|
| 'scintillation', 'fanon', 'rubato', 'upstanding', 'collard', 'creeks', 'skookum', 'unbelievers', 'monocyte', 'nishikawa', 'crusher', 'gerwen', 'abrah', 'silverchair', 'hangman', 'unitary', 'klausen', 'arousal', 'heat', 'bridgnorth', 'mildred', 'porton', 'aquasox', 'wylie', 'hipaa', 'krimuk', 'hexahedron', 'kuei', 'barbera', 'dalvi', 'gilding', 'visakhapatnam', 'tatsuo', 'tarascon', 'bajram', 'scholes', 'hadad', 'incidental', 'theodosius', 'reichskommissariat', 'boeheim', 'amsl', 'buencamino', 'thrasyvoulos', 'insulated', 'discourtesy', 'nisra', 'ycko', 'luen', 'dooku' | 'nine', 'ranked', 'zero', 'the', 'garcia', 'rank', 'station', 'the', 'for', 'four', 'williams', 'drunken', 'a', 'one', 'eight', 'of', 'were', 'zero', 'debate', 'orchestra', 'of', 'wrist', 'points', 'fractured', 'the', 'to', 'redirect', 'adnan', 'white', 'car', 'fond', 'concluded', 'under', 'two', 'by', 'five', 'his', 'infection', 'the', 'the', 'pop', 'in', 'one', 'in', 'one', 'one', 'fram', 'handled', 'battle', 'mutual' |

## 3 Embedding symmetry

### 3.1 Definition of embedding symmetry

In mathematical science fields, such as high-dimensional probability theory [55] and the volume of convex bodies [27], there are numerous intriguing definitions of spatial symmetry. Among them, we begin with the definition of the symmetry of *random* vectors with their frequencies [48, 55]. This is suited for dealing with word vectors because they entail word *frequencies*, unlike usual data instances.

> **Definition 1** (A random vector $\boldsymbol{v} \sim p$ on $\mathbb{R}^d$ has zero mean; the *1st* moment of a symmetric random vector).
> $$\overline{\boldsymbol{v}} := \mathbb{E}_{\boldsymbol{v} \sim p}[\boldsymbol{v}] = \boldsymbol{0} \qquad (1)$$

> **Definition 2** (A random vector $\boldsymbol{v} \sim p$ on $\mathbb{R}^d$ is in isotropic position around its barycenter; the *2nd* moment of a symmetric random vector).
> $$\mathrm{Cov}[\boldsymbol{v}] := \mathbb{E}_{\boldsymbol{v} \sim p}[(\boldsymbol{v} - \mathbb{E}_{\boldsymbol{v} \sim p}[\boldsymbol{v}])(\boldsymbol{v} - \mathbb{E}_{\boldsymbol{v} \sim p}[\boldsymbol{v}])^\top] \propto \boldsymbol{I}_d \ (2)$$

From these definitions, we will develop methods to *adjust* given word vectors to be symmetric in § 3.2, and to *evaluate* the symmetry of given word vectors in § 3.3.

In machine learning and NLP, the spatial symmetry of embedding spaces is a hot topic, and numerous theories and algorithms have been proposed [41, 21, 38, 56]. However, the approach in many researches implicitly treats all vectors equally, ignoring word frequency information. In the following sections, we will detail both the empirical and theoretical issues that a uniform approach can cause, especially when applied to NLP tasks. Furthermore, when embeddings correspond to tokens rather than types—such as in the internal representations of masked or causal language models—a uniform approach tends to be effective. This point will be discussed in § 5.1.

### 3.2 Enhancement of embedding symmetry

This section proposes **Zipfian whitening**[2], which symmetrizes a given set of word vectors with word frequency. At a glance, the most natural method to achieve Def. 1 and Def. 2 would be PCA whitening, also known as sphering. Notably, each step of whitening—centering, decorrelation, and standardization—implicitly involves calculating expected values. Our approach is simple: each time we calculate an expected value, we should weight it by the empirical word frequency. The specific algorithm is as shown in Algorithm 1. The only difference from general whitening is that it uses word frequency in the part highlighted in blue . Please refer to Appendix A for a formal explanation showing that the word vectors obtained by the proposed algorithm actually satisfy Def. 1 and Def. 2.

---

[2]In this paper, "Zipfian" is simply used to denote a "highly non-uniform" distribution. Our focus is on the mismatch between actual word frequencies and uniform distribution, and we have *not* constructed arguments or experiments that rely on specific properties of power laws. Refining experiments and theory based on the degree of tail heaviness is an interesting direction for future work.

---

**Algorithm 1** Zipfian whitening; a post-processing algorithm on word embeddings. The part high-lighted in blue shows the difference from the typical centering and whitening.

---

**Input:** Word embeddings $\{\boldsymbol{w}_i \in \mathbb{R}^d \mid w_i \in \mathcal{V}\}$, word frequency $p\colon \mathcal{V} \to [0,1]$.

**Output:** Processed word embeddings. $\{\overline{\boldsymbol{w}}_i \in \mathbb{R}^d\}_i$ are centered, $\{\widetilde{\boldsymbol{w}}_i \in \mathbb{R}^d\}_i$ are further whitened.

   *Zipfian centering (1st moment):*
1: $\widehat{\boldsymbol{\mu}} \leftarrow \sum_{w_i \in \mathcal{V}} p(w_i) \boldsymbol{w}_i \in \mathbb{R}^d$
2: **for all** $w_i \in \mathcal{V}$ **do**
3: $\quad \overline{\boldsymbol{w}}_i \leftarrow \boldsymbol{w}_i - \widehat{\boldsymbol{\mu}} \in \mathbb{R}^d$
4: **end for**

   *Zipfian decorrelation and standardization (2nd moment):*
5: $\boldsymbol{W}_p \leftarrow \left[ \sqrt{p(w_1)}\,\overline{\boldsymbol{w}}_1^\top, \ldots, \sqrt{p(w_{|\mathcal{V}|})}\,\overline{\boldsymbol{w}}_{|\mathcal{V}|}^\top \right]^\top \in \mathbb{R}^{|\mathcal{V}| \times d}$
6: $\boldsymbol{U}\boldsymbol{\Sigma}\boldsymbol{V}^\top \leftarrow \mathrm{SVD}(\boldsymbol{W_p})$
   $\qquad \triangleright \ \boldsymbol{\Sigma} = \mathrm{diag}(\sigma_1, \ldots, \sigma_d) \in \mathbb{R}^{d \times d}$ consists of the singular values of $\boldsymbol{W}_p$.
7: **for all** $w_i \in \mathcal{V}$ **do**
8: $\quad \widetilde{\boldsymbol{w}}_i \leftarrow \overline{\boldsymbol{w}}_i \boldsymbol{V} \boldsymbol{\Sigma}^{-1}$
   $\qquad \triangleright \ \boldsymbol{\Sigma}^{-1} := \mathrm{diag}(1/\sigma_1, \ldots, 1/\sigma_d) \in \mathbb{R}^{d \times d}$.
9: **end for**

---

Table 2: The empirical performance of Zipfian whitening, which exploits the empirical frequency of words during expectation calculations. Each cell shows the STS-B [15] score $\times 100$. By carefully performing the simple operation of whitening, it consistently outperforms powerful baseline methods.

| GloVe | 46.17 | | Word2Vec | 56.98 | |
|---|---|---|---|---|---|
| | Uniform | Zipfian | | Uniform | Zipfian |
| + Centering | 45.17 | **52.25** | + Centering | 55.85 | **58.84** |
| + Whitening | 52.21 | **66.92** | + Whitening | 56.03 | **66.50** |
| + ABTT [39] | 54.28 | | + ABTT [39] | 56.98 | |
| + SIF + CCR [7] | 58.70 | | + SIF + CCR [7] | 63.04 | |

**Empirical evaluation:** We confirm the effectiveness of Zipfian whitening (Algorithm 1) by measuring performance on standard sentence-level downstream tasks using post-processed word vectors. We employed the most standard word embeddings—GloVe [43], word2vec [37], and fastText [11]—and utilized the widely adopted evaluation tasks, including STS-B [15] and related benchmarks. Detailed experimental settings can be found in Appendix B. Table 2 shows the results on the STS-B task. Remarkably, the proposed Zipfian whitening shows significant advantages not only over standard (uniform) centering and whitening but also over the strong baseline method [7] specifically designed to create powerful sentence vectors. Consistent results were obtained with various benchmark datasets, multiple empirical word probabilities, and a language other than English (Appendix C)[3]. In § 4.2.1, one reason for this remarkable performance is clarified from the perspective of information geometry.

## 3.3 Evaluation of embedding symmetry

The community is greatly interested not only in making word vector spaces symmetric but also in evaluating *how symmetric* or asymmetric a space is [21, 49]. Here, we return to Def. 1 and Def. 2 and describe metrics for evaluating the symmetry of word embedding spaces with word frequency.

**Degree of centrality—the 1st moment of symmetry:** Recall that, if the barycenter $\mathbb{E}[\boldsymbol{v}]$ is close to $\boldsymbol{0}$, then the random vector $\boldsymbol{v}$ can be considered symmetric in terms of the first moment (Def. 1).

---

[3]Notably, we observed improved scores when using word frequencies from the evaluation dataset itself as $p(w)$. In general, for NLP tasks, $p(w)$ refers to word frequencies derived from the embedding training data or from a standard large corpus. However, to optimize downstream task performance, it is preferable to base $p(w)$ on word frequencies within the evaluation dataset itself used for those tasks. This adjustment exemplifies "covariate shift" [52] in machine learning, where the distribution of training data differs from that of test data.

Thue, examining the value of $\|\mathbb{E}[\boldsymbol{v}]\| := \mathbb{E}[\boldsymbol{v}] - \boldsymbol{0}$ appears to be a reasonable way to measure the symmetry of the first moment. However, random vectors $\boldsymbol{v}$ and $\alpha\boldsymbol{v}$ ($\alpha \in \mathbb{R}_{>0}$) should be considered equivalent in terms of spatial symmetry. Thus, we define the scale-invariant metric (Def. 3), obtained by dividing $\|\mathbb{E}[\boldsymbol{v}]\|$ by the average length $\mathbb{E}[\|\boldsymbol{v}\|]$.

---

**Definition 3** (Degree of centrality for the random vector $\boldsymbol{v} \sim p$; the 1st moment of symmetry)**.**

$$\mathrm{Sym}_1(\boldsymbol{v}) := 1 - \left\| \underset{\boldsymbol{v}\sim p}{\mathbb{E}}[\boldsymbol{v}] \right\| / \underset{\boldsymbol{v}\sim p}{\mathbb{E}}[\|\boldsymbol{v}\|] \tag{3}$$

---

By definition, $\mathrm{Sym}_1(\boldsymbol{v})$ takes values in $[0, 1]$, and $\mathrm{Sym}_1(\boldsymbol{v}) = 1$ if and only if $\boldsymbol{v}$ is zero mean.

**Degree of isotropy—the 2nd moment of symmetry:** If the covariance matrix $\mathbb{E}[(\boldsymbol{v} - \mathbb{E}[\boldsymbol{v}])(\boldsymbol{v} - \mathbb{E}[\boldsymbol{v}])^\top]$ is a constant multiple of the identity matrix $\boldsymbol{I}_d$, i.e., if the random vector $\boldsymbol{v}$ has an equal spread in all directions, $\boldsymbol{v}$ is symmetric in terms of the second moment (Def. 2). Following convention, this degree can be confirmed by examining the flatness of the eigenspectrum.

---

**Definition 4** (Degree of isotropy around the barycenter for the random vector $\boldsymbol{v} \sim p$; the 2nd moment of symmetry)**.**

$$\mathrm{Sym}_2(\boldsymbol{v}) := \frac{1}{\log d} H\left( \frac{\lambda_1}{\sum_j \lambda_j}, \dots, \frac{\lambda_d}{\sum_j \lambda_j} \right) \tag{4}$$

$\{\lambda_1, \dots, \lambda_d\}$ *are the eigenvalues of the covariance matrix* $\underset{\boldsymbol{v}\sim p}{\mathbb{E}}[(\boldsymbol{v} - \underset{\boldsymbol{v}\sim p}{\mathbb{E}}[\boldsymbol{v}])(\boldsymbol{v} - \underset{\boldsymbol{v}\sim p}{\mathbb{E}}[\boldsymbol{v}])^\top]$. $H(p_1, \dots, p_d) := -\sum_i p_i \log p_i$ *is the Shannon entropy.*

---

**Proposition 1.** $\mathrm{Sym}_2(\boldsymbol{v})$ *takes values in* $[0, 1]$*, and* $\mathrm{Sym}_2(\boldsymbol{v}) = 1$ *if and only if* $\boldsymbol{v}$ *is isotropic around its barycenter (Def. 2).*      *Proof.* Please refer to Appendix D.

Note that the approach of measuring the entropy of the spectrum to evaluate the flatness of a signal can be found in many fields. For example, similar definitions are seen in probability processes [14] and signal processing [17, 47]. We also follow this standard and powerful line.

**Algorithm:** To compute the evaluation metrics of symmetry (Def. 3, Def. 4) for given word vectors, again, one should just use the empirical word frequency when calculating the expectations. A pseudocode for measuring symmetry is provided in Appendix E.

**Empirical evaluation:** To what extent does our symmetry score (an intrinsic evaluation of embedding spaces) correlate with downstream task performance (an extrinsic evaluation of those)? As baselines, we use versions of our symmetry score that do *not* account for word frequency, calculated in a uniform manner. We also compare with popular symmetry scores in NLP, the average of cosine similarity (**Ave. Cos.**) [21] and the recently proposed **IsoScore** [49]. Note that all these baselines implicitly assume uniform word frequency. Additional experimental settings can be found in Appendix B. Fig. 2 shows the results. The right side of Fig. 2 demonstrates the superiority of the Zipfian approach. Moving from the bottom-left to the top-right of the figure—i.e. as both the 1st ($x$-axis) and 2nd moments ($y$-axis) of the symmetry score increase—it is clearly visible that the downstream task performance increases (the color becomes more red). In contrast, in the left-hand plot, which assumes uniform word frequency, there is no observed relationship between the symmetry score ($x$ and $y$-axis) and the downstream task performance (color). Table 3 lists the correlation coefficients between the symmetry scores and downstream task performance in more detail. It can be seen that the symmetry scores considering word frequency can "predict" downstream task performance with remarkably high correlation. On the other hand, the "prediction" performance of other metrics, including Ave. Cos. and IsoScore that implicitly assume uniform word frequency, is unsatisfactory. Surprisingly, when the most popular Ave. Cos. metric shows almost no correlation (0.04) with downstream task performance (STS-B), Zipfian symmetry metric has a strong positive correlation (0.83) with it.

## 4  Why is Zipfian whitening better than uniform whitening?

A natural question is why the Zipfian approach empirically dramatically outperforms the uniform approach. We provide a theoretical explanation using Table 4. In a nutshell, a significant difference arises depending on whether the base measure of an exponential family is uniform or Zipfian.

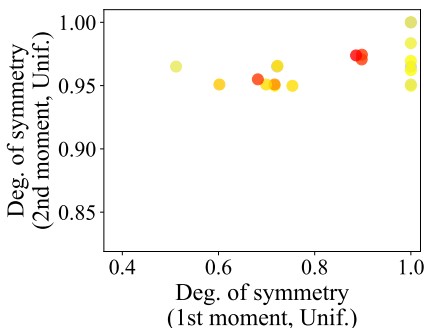 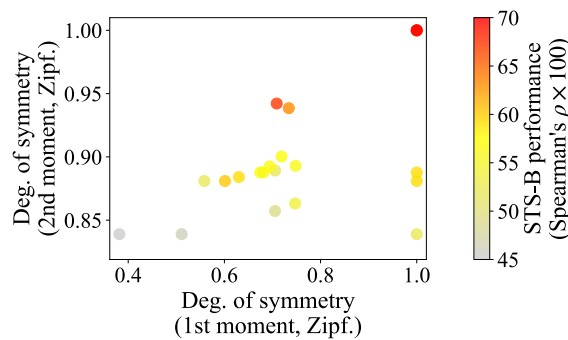

Figure 2: The relationship between the 1st-order symmetry (Def. 3, $x$-axis), the 2nd-order symmetry (Def. 4, $y$-axis), and task performance (color). Each point represents either pre-trained or post-processed word embeddings (GloVe, word2Vec, and fastText). The Zipfian measure well captures the downstream task performance (**right**), while the uniform isotropic measure cannot (**left**).

Table 3: Spearman's $\rho \times 100$ (each cell) between the symmetry scores (each column) and downstream STS-B performance (each row), on pre-trained and post-processed embeddings (GloVe, word2Vec, and fastText). The scores based on the Zipfian prior show a significantly higher correlation with task performance compared to those based on the uniform prior including Ave. Cos. and IsoScore.

|  | Ave. Cos. [21] | IsoScore [49] | Uniform | | Zipfian | |
|---|---|---|---|---|---|---|
|  |  |  | 1st moment | 2nd moment | 1st moment | 2nd moment |
| STS-B | $-6.95$ | $0.07$ | $-21.91$ | $-21.21$ | $62.13$ | **89.55** |
| SICK-R | $20.09$ | $18.41$ | $13.26$ | $-8.71$ | $60.04$ | **64.60** |

## 4.1 Characterization through generative model, partition function, and whitening

**Exponential families:** Hereafter, we interpret two salient generative models from the viewpoint of *exponential families*: one given by Arora et al. [6] and the other generalizing the Levy–Goldberg formula [32, Eq. (7)]. Details of these models will be provided shortly. An exponential family is a class of probability distributions of a random variable $\boldsymbol{x}$ parametrized by a parameter $\boldsymbol{\theta}$, written in the following (canonical) form:

$$p(\boldsymbol{x} \mid \boldsymbol{\theta}) = \pi(\boldsymbol{x}) \exp\left(\langle \boldsymbol{x}, \boldsymbol{\theta} \rangle - \psi(\boldsymbol{\theta})\right), \quad \psi(\boldsymbol{\theta}) \coloneqq \log Z(\boldsymbol{\theta}) = \log\left(\sum_{\boldsymbol{x}} \pi(\boldsymbol{x}) \exp(\langle \boldsymbol{x}, \boldsymbol{\theta} \rangle)\right), \quad (18)$$

where $\boldsymbol{x}$ is a sufficient statistic, $\boldsymbol{\theta}$ is called a natural parameter, $\pi$ is the *base measure* (or "prior"), and $\psi$ is the log-partition function. Once we specify the base measure $\pi$ and the canonical pair $(\boldsymbol{x}, \boldsymbol{\theta})$, the log-partition function is determined. That being said, the base measure $\pi$ is the design choice of an exponential family left for us. In the following, we specifically examine an exponential family of distributions in the form $p(w \mid c)$, where word $w$ is predicted given context $c$. Specifically, the *context* represents a co-occurring word (in static word embeddings), a cloze sentence (in masked language models), or a sentence prefix (in causal language models). In all of these cases, we predict a word with the logit $\langle \boldsymbol{w}, \boldsymbol{c} \rangle$, making the exponential family a natural probabilistic model. Here, the vector $\boldsymbol{c}$ represents the vector expression of the context $c$, known as the "context vector." Note that, even for the same word $t$, the predicted word vector $\boldsymbol{w}(t)$ and the predicting context vector $\boldsymbol{c}(t)$ are distinct.

**Uniform prior:** Arora et al. firstly considered a log-linear generative model of word embeddings given a context (6) and demonstrated that when the generative model is adopted with normalized context vectors and a huge vocabulary, the partition function asymptotically becomes constant (8) [6, Lemma 2.1]. Here, we can regard that this model belongs to the exponential family with the uniform base measure $\pi(w) = \pi(c) = 1/|\mathcal{V}|$ [4].

---

[4]Although Arora et al. [6]'s generative model treats a context vector $\boldsymbol{c}$ as a model parameter drifting by a random walk, we can cast their model into an exponential family because they did not specify how the initial context vector is generated. Hence, by regarding $c$ as an observed token with the uniform prior $\pi(c) = 1/|\mathcal{V}|$,

Table 4: Through the differences in the underlying generative models, the empirical superiority of Zipfian whitening over uniform whitening can be understood.

---

**Generative models** behind the (whitened) embeddings

$$p(w \mid c) = \frac{\pi(w)\exp(\langle \boldsymbol{w}, \boldsymbol{c}\rangle)}{Z(c)}, \quad Z(c) = \sum_w \pi(w)\exp(\langle \boldsymbol{w}, \boldsymbol{c}\rangle); \quad p(w, c) = p(w \mid c)\pi(c) \tag{5}$$

| with **Uniform prior** | with **Zipfian prior** |
|---|---|
| $p_{\textcircled{u}}(w \mid c) = \dfrac{1\exp(\langle \boldsymbol{w}, \boldsymbol{c}\rangle)}{Z_{\textcircled{u}}(c)}, \pi(c) \propto 1$ (6) | $p_{\textcircled{z}}(w \mid c) = \dfrac{p(w)\exp(\langle \boldsymbol{w}, \boldsymbol{c}\rangle)}{Z_{\textcircled{z}}(c)}, \pi(c) = p(c)$ (7) |

**Partition functions** become constant under certain conditions

| Assume $\|\boldsymbol{c}\| \equiv 1$, $|\mathcal{V}| \to \infty$, then | At the optimal solution of the corresponding loss, |
|---|---|
| $Z_{\textcircled{u}}(c) := \sum_w \exp(\langle \boldsymbol{w}, \boldsymbol{c}\rangle) = \text{const.}$ (8) [6] | $Z_{\textcircled{z}}(c) := \sum_w p(w)\exp(\langle \boldsymbol{w}, \boldsymbol{c}\rangle) = \text{const.}$ (9) [32] |

**Whitening** coarsely achieves a constant partition function

| $Z_{\textcircled{u}}(c)$ | $Z_{\textcircled{z}}(c)$ |
|---|---|
| $= |\mathcal{V}| + \left(\underbrace{\sum_w \boldsymbol{w}}\right)^{\top}\boldsymbol{c} + \frac{1}{2}\boldsymbol{c}^{\top}\left(\underbrace{\sum_w \boldsymbol{w}\boldsymbol{w}^{\top}}\right)\boldsymbol{c} + \dots$ | $= |\mathcal{V}| + \left(\underbrace{\sum_w p(w)\boldsymbol{w}}\right)^{\top}\boldsymbol{c} + \frac{1}{2}\boldsymbol{c}^{\top}\left(\underbrace{\sum_w p(w)\boldsymbol{w}\boldsymbol{w}^{\top}}\right)\boldsymbol{c} + \dots$ |
| $\rightsquigarrow |\mathcal{V}| + \underbrace{\boldsymbol{0}}^{\top}\boldsymbol{c} \quad + \frac{1}{2}\boldsymbol{c}^{\top}\underbrace{\boldsymbol{I}}\boldsymbol{c}\dots \approx \text{const.}$ (10)[39] | $\rightsquigarrow |\mathcal{V}| + \underbrace{\boldsymbol{0}}^{\top}\boldsymbol{c} \quad + \frac{1}{2}\boldsymbol{c}^{\top}\underbrace{\boldsymbol{I}}\boldsymbol{c}\dots \approx \text{const.}$ (11)[ours] |

**Vector norm** under generative models

| $\|\boldsymbol{w}\|_2^2 \approx 2d\log p(w) - 2Z$ (12) [6] | $\|\boldsymbol{w}\|_{\boldsymbol{G}(w)}^2 \approx 2\text{KL}(p(\cdot)\|p(\cdot \mid w))$ (13) [42] [Thm. 1] |
|---|---|
| long vector $\leftrightarrow$ frequent (*un*informative) word | long vector $\leftrightarrow$ informative word |

**Loss** and **error** corresponding to generative models $p(w \mid c)$

| softmax cross-entropy loss | logit-adjusted softmax cross-entropy loss |
|---|---|
| $\underset{(w,c)}{\mathbb{E}} -\log \dfrac{\exp(\langle \boldsymbol{w}, \boldsymbol{c}\rangle)}{Z_{\textcircled{u}}(c)}$ (14) | $\underset{(w,c)}{\mathbb{E}} -\log \dfrac{p(w)\exp(\langle \boldsymbol{w}, \boldsymbol{c}\rangle)}{Z_{\textcircled{z}}(c)}$ (15) [36] |

| misclassification error | balanced error |
|---|---|
| $\mathbb{P}_{(w,c)}[w \notin \underset{w'}{\arg\max}\langle \boldsymbol{w}', \boldsymbol{c}\rangle]$ (16) | $\frac{1}{|\mathcal{V}|}\sum_{w \in \mathcal{V}} \mathbb{P}_{c|w}[w \notin \underset{w'}{\arg\max}\langle \boldsymbol{w}', \boldsymbol{c}\rangle]$ (17) [36] |

---

**Zipfian prior:** An exponential family adopted with the Zipfian measure can be written as (7). This generative model can be naturally derived from the skip-gram model with negative sampling (SGNS) [37]. By assuming that the linear model $c \mapsto \langle \boldsymbol{w}, \boldsymbol{c}\rangle$ is sufficiently capable of discriminating cooccurring words and negative samples (as in the realizable case), we can see that the generative model of the word embeddings must comply with the following formula:

$$\log \frac{p(w, c)}{p(w)p(c)} - \log k = \langle \boldsymbol{w}, \boldsymbol{c}\rangle, \tag{19}$$

where $k$ is the number of negative samples. This optimality formula owes to Levy and Goldberg [32], and we call (19) the *Levy–Goldberg formula*. A more concise derivation is later given by Oyama et al. [42]. We can regard the Levy–Goldberg formula as an exponential family with the Zipfian base measure, $\pi(w) = p(w)$ and $\pi(c) = p(c)$, and the constant log-partition function $Z_{\textcircled{z}}(c) \equiv k^{-1}$. The generative model (7) is a relaxation of the Levy–Goldberg formula since we do not impose the realizability assumption necessary for the derivation of (19).

---

their model is reduced to (6). The static context prior does not contradict Arora et al. [6]'s model with sufficiently large $d$, where the random walk drifts extremely slowly.

**What does whitening do?** Mu and Viswanath [39] proposed a method to approximately make the partition function of the uniform prior model constant by centering the word vectors and removing the top principal components (10). Our Zipfian whitening corresponds to Mu and Viswanath's post-processing method, in the sense that ours and theirs make the partition function constant up to the second moment (11) and (10), respectively. In summary, Zipfian whitening (11) transforms a probabilistic model into an exponential family adopted with the Zipfian base measure (7), making it closer to the Levy–Goldberg formula (19).

## 4.2 Emphasis on rare words by Zipfian prior

Let us explore further why the Zipfian prior results in good performance in downstream tasks (§ 3.2). In summary, the Zipfian prior approach emphasizes low-frequency words, while the uniform prior approach emphasizes high-frequency words, both from perspectives of vector norms and errors/losses. So far in this paper, we have repeatedly discussed weighting each word according to frequency, so it may seem contradictory that Zipfian approach emphasizes low-frequency words as a result. To illustrate, let us reconsider centering. In centering, the mean vector is *subtracted* from each vector. Weighting each word vector by frequency when constructing the mean vector means that signals corresponding to high-frequency words are removed more substantially from each vector. The emphasis on low-frequency words has been repeatedly supported throughout the history of NLP and information retrieval, such as Luhn's hypothesis [34], inverse document frequency (IDF) [53], and smooth inverse frequency (SIF) [7]. For instance, it is reasonable to emphasize the word 'isotropy' when creating a sentence embedding containing both words 'the' and 'isotropy'.

### 4.2.1 From the perspective of vector norm

Under the Zipfian prior model, *words with larger information content have longer (emphasized) vector representations*. Conversely, under the uniform prior model, words with smaller information content have longer (emphasized) vector representations.

As a representative example of **uniform prior** models, the norms of word vectors learned by random walk language models are theoretically and empirically proportional to word frequency (12) (see Eq. (2.4) and Fig. 2 in Arora et al. [6]). That is, in such embedding space, words with *less* information (e.g., 'the') are emphasized. This tendency is consistently observed in dynamic language models and causal language models that adopt the softmax cross-entropy loss, another typical example of the uniform prior family [28]. By contrast, when training word embeddings with skip-gram negative sampling [37], the word embeddings follow the **Zipfian prior** family, and their norms become larger with greater information, which we show subsequently [50, 60, 42]. Based on the formulation of the exponential family and following Eq. (12) of Oyama et al. [42], we formally describe the norm properties of the word vectors obtained from the Zipfian prior model.

**Theorem 1** (The norm of a word vector learned with empirical Zipfian prior models reflect the information amount of the word; a refined version of [42] Eq. (12)). *Assume that word embeddings $\{w_i\}_i$ follow the Zipfian prior model (7), For the same word $t$, the vector $\boldsymbol{w}$ on the predicted side and the vector $\boldsymbol{c}$ on the predicting side are shared: $\boldsymbol{w}(t) \equiv \boldsymbol{c}(t)$ (weight tying), and $\sum_{t \in \mathcal{V}} p(t)\boldsymbol{w}(t) = \boldsymbol{0}$ (centered w.r.t. Zipfian prior), then each word vector $\boldsymbol{w}(t)$ satisfy*

$$\|\boldsymbol{w}(t)\|^2_{\boldsymbol{G}(t)} \approx 2\mathrm{KL}(p(\cdot)\|p(\cdot \mid t)), \quad \boldsymbol{G}(t) := \sum_{t' \in \mathcal{V}} p(t' \mid t)\boldsymbol{c}(t')\boldsymbol{c}(t')^\top, \tag{20}$$

*where $\|\boldsymbol{w}\|_{\boldsymbol{A}}$ with a positive definite matrix $\boldsymbol{A}$ denote a norm based on a quadratic form $\sqrt{\boldsymbol{w}^\top \boldsymbol{A} \boldsymbol{w}}$[5].*
*Proof.* Refer to Appendix F.

In Fig. 3, we experimentally confirmed that the norms of informative words become larger with Zipfian whitening (shown from center to the right in Fig. 3), bringing them closer to the ideal Zipfian prior model[6].

---

[5]The matrix $\boldsymbol{G}(w)$ takes the form $\sum_i \boldsymbol{x}_i \boldsymbol{x}_i^\top$ is indeed positive definite, similar to a covariance matrix.

[6]Given these results, some readers may be interested in how the Zipfian whitening process influences the lengths and directions of vectors individually. For details of the ablation study, please refer to Appendix G.

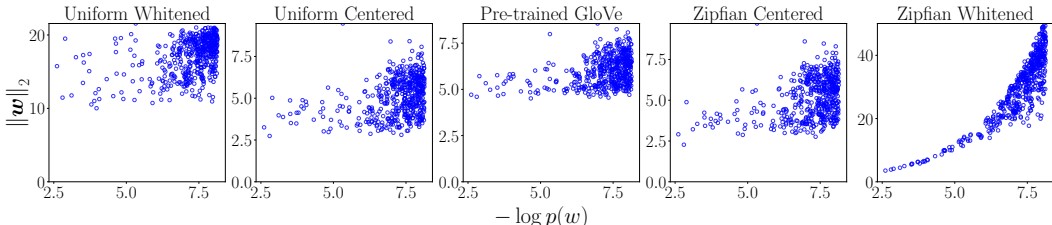

Figure 3: Relationships between the information content $-\log p(w)$ and the vector norms $\|\boldsymbol{w}\|_2$ for top 500 frequent words $w$. The figure in the center represents the pre-trained GloVe model. By using **Zipfian** whitening, the information content gets encoded in the norm (center to right). Conversely, with **uniform** whitening, this phenomenon does not occur (center to left).

#### 4.2.2 From the perspective of error and loss

*The error and loss functions associated with the Zipfian prior model emphasize low-frequency words.* In contrast, the error and loss functions of the uniform prior model focus on the average loss across the entire dataset, resulting in a greater emphasis on high-frequency words.

The standard classification loss is the softmax cross-entropy loss (14). By taking its expectation over the dataset $\{(w, c)\}$, embeddings associated with higher-frequency words receive more updates because the softmax is the uniform inverse link, corresponding to the **uniform prior** model. By contrast, the logit-adjusted loss (15) has been proposed to tackle class imbalance [36]. From our viewpoint, the logit adjustment term $p(w)$ makes the inverse link belong to the **Zipfian prior** model. The softmax and logit-adjusted losses are Fisher consistent to the misclassification (16) and balanced (17) error rates, respectively. As the latter tends to stress minor classes, the logit-adjusted loss and Zipfian prior model are suitable for emphasizing low-frequency words during the learning process.

Another prominent loss function for representation learning is contrastive loss, with the SGNS loss (word2vec) [37] as a representative example in the context of word representation learning. This loss similarly uses a loss aligned with the **Zipfian prior**:

$$- \mathop{\mathbb{E}}_{(w,c)}\left[\log \sigma(\langle \boldsymbol{w}, \boldsymbol{c}\rangle) + \sum_{\substack{i=1,\ldots,k \\ w'_i \sim p(w)}} \log \sigma(-\langle \boldsymbol{w}'_i, \boldsymbol{c}\rangle)\right], \tag{21}$$

where $\sigma$ is sigmoid function, and $k$ is the number of negative samples. Since high-frequency words are more likely to be sampled as negative examples, the loss has less impact on high-frequency words in positive examples. Consequently, low-frequency positive words are relatively emphasized in representation learning. The Levy–Goldberg formula in the previous section describes the properties of an ideally trained word2vec model, which are essentially the properties of Zipfian prior models.

## 5 Unified explanation of the efficacy of existing methods

Distinguishing the distribution that the base measure follows helps us understand why some existing NLP methods are effective.

### 5.1 Uniform whitening of token embeddings $\approx$ Zipfian whitening of type embeddings

Masked language models like BERT [18] and RoBERTa [33] produce dynamic (contextualized) *token* embeddings. Adding up such token embeddings of constituent tokens to create sentence embeddings often leads to poor empirical performance [46]. However, symmetrizing significantly improves their performance; such methods including "batch *centering*," "*Whitening*BERT," and contrastive learning methods [16, 46, 59, 22, 26, 57]. This improvement can also be explained from the perspective of the Zipfian prior. A dataset or corpus is first fed into the model to obtain *token* embeddings[7]. Centering/whitening is then applied to this entire set of embeddings. As this token embedding (multi)set has the multiplicity asymptotically proportional to the word frequency,

---

[7]Here, the computation of the additive composition $\boldsymbol{s} := 1/|s| \sum_{w \in s} \boldsymbol{w}$ can be ignored without major issues in formal discussions of spatial symmetry. This is because the words in a sentence are generated based on word

Table 5: The empirical performance difference between "uniform"—enforced centering and whitening with a uniform prior for dynamic embeddings, and "Zipfian"—conventional uniform centering and whitening over tokens with an implicit Zipfian prior over types. Each cell shows the STS-B [15] score $\times 100$. This comparison reveals that token-level uniform centering/whitening, corresponding to type-level Zipfian centering/whitening, leads to empirically better performance.

| BERT-base | 63.75 | | | RoBERTa-base | 60.75 | |
|---|---|---|---|---|---|---|
| | "Uniform" | "Zipfian" | | | "Uniform" | "Zipfian" |
| + Centering | 64.04 | **64.82** | | + Centering | 60.34 | **61.30** |
| + Whitening | 60.53 | **64.91** | | + Whitening | 61.31 | **65.59** |

this *uniform* centering/whitening of *token* embeddings corresponds to the word-frequency-weighted (*Zipfian*) centering/whitening of *type* embeddings. For a more formal description of the above explanations, please refer to Appendix H. Additionally, recent work has found that contrastive additive sentence encoders implicitly weight words by their information content [30]. This finding is consistent with the previous discussion on vector norms (§ 4.2.1), and can be seen as indirect evidence supporting the idea that these models belong to the Zipfian prior family.

This idea can also be supported by empirical evidence. This idea is also supported by empirical evidence. To establish a baseline for centering and whitening token embeddings under a uniform prior, we scale each embedding by the reciprocal of its type frequency, ensuring uniform treatment across types. Refer to the Appendix H for the detailed computation of this *pseudo uniform* approach and a formal explanation of how it achieves type uniformity. Table 5 shows the results. Comparing the pseudo-uniform centering/whitening (which assumes a *uniform* prior over types) with the conventional token-level uniform centering/whitening (which implicitly assumes a *Zipfian* prior over types) reveals that the latter approach based on a Zipfian prior empirically achieves better performance. Additional experimental settings and results can be found in Appendix B and Appendix I.

## 5.2 Headless causal language model roughly belongs to Zipfian prior family

The recently proposed headless language model [23] uses only words within the same batch to predict next tokens with a pseudo-softmax function. This method originally aimed to reduce the computational cost of the softmax function in the $|\mathcal{V}|$ direction, but an interesting side effect is the improvement in the performance. This success can also be explained from the perspective of Zipfian priors. If we repeatedly sample small batches, the sampling frequency of each word will increasingly reflect its true frequency as the batch size approaches 1.

## 6 Conclusion

Standard methods for adjusting and measuring symmetries in word embedding spaces—such as centering and whitening—implicitly assume *uniformly* distributed word frequencies, which is unrealistic. We hypothesize that, based on the type-token distinction, using empirical *Zipfian* word frequencies is essential when calculating the expectation (§ 2). Based on the idea and the definitions of first- and second-order symmetry in random vectors, we derived Zipfian whitening, which *enhances the symmetry* of the word embedding space. Even though it is nearly identical to standard PCA whitening, Zipfian whitening significantly outperforms existing methods (§ 3.2). Similarly, we derived a metric to *evaluate the symmetry* of word embedding spaces. Our intrinsic metrics showed a strong correlation with extrinsic task performance, even when popular metrics show almost none (§ 3.3). We then presented a framework explaining the differences in effect between whitening based on uniform and Zipfian approaches, by attributing them to differences in the base measure of the exponential family (§ 4.1). By further exploring this viewpoint through information geometry and loss functions, we showed how the Zipfian approach emphasizes the informativeness of low-frequency words (§ 4.2.1). Lastly, through our proposed viewpoint, we found that popular NLP methods perform well because their word embeddings end up encoding a Zipfian prior; such models include word2vec [37] (§ 4.2.2), WhiteningBERT [26] (§ 5.1), and headless language models [23] (§ 5.2).

---

frequency distribution, resulting in the first and second moments (Def. 1, Def. 2) of sentence vectors closely matching those of word vectors.

## Acknowledgements

This work is supported by JST ACT-X Grant Number JPMJAX200S and JSPS KAKENHI Grant Number 22H05106. We received numerous constructive and valuable comments from the anonymous reviewers of NeurIPS 2024, which have significantly contributed to the quality improvements from the submission version to the camera-ready version. We would also like to thank Hayato Tsukagoshi of Nagoya University for his insightful comments on the handling of dynamic embeddings and on the experimental setup of the SimCSE paper [22], including minor discrepancies between the paper's description and its actual implementation (also see Footnote 16). We also extend our gratitude to the organizers and participants of MLSS 2024, the Tohoku NLP group, the Shimodaira lab at Kyoto University, and many members of the Japanese NLP and machine learning community, for their constructive feedback and motivating encouragement throughout our research discussions.

## Limitations

### How these assumptions might be violated in practice

In our theoretical analysis concerning norms, and in the discussion on the relationship between whitening and normalization constants, we have proceeded by ignoring the residual terms beyond the second order. Empirically, focusing only on the first and second order has yielded significant results. However, to accurately identify cases where the proposed method might fail, a detailed theoretical and empirical examination of the asymptotic behavior of higher-order moments might be crucial. This remains an important future work.

The condition that the partition function is constant is only a necessary condition from the perspective of both the generative model's optimal solution and whitening. The true logical relationship between whitening and the generative model has not been clarified. In particular, verifying whether the projection through whitening allows us to transition between the two model families (the uniform family and the Zipfian family) is an intriguing and valuable direction for both theoretical exploration and practical application.

### The scope of the empirical claims made

Our experiments primarily focused on static and dynamic word embeddings, as many of their theoretical properties have been understood and they have been central to the rise of isotropization. Admittedly, this paper also advances our understanding of causal language models. However, to make a more significant practical impact in the era of large language models, employing the proposed method as a regularization term for next-token prediction holds great promise for future work.

The experiments utilized typical downstream NLP tasks, particularly popular datasets for sentence-level semantic tasks. By scaling up the task set to include word-level tasks or leveraging a broader range of multilingual data, we can more robustly demonstrate the practical utility of the proposed framework.

### The factors that influence the performance of our approach

The proposed method inherently involves numerically unstable calculations, such as multiplying by the inverse of small singular values. Embeddings for low-frequency words are often far from converged even after extensive pre-training, and the eigenvalues of the embedding space are known to decay. Given these situations, the adverse effects of small singular values are plausible. Considering recent advancements in whitening techniques, developing a more numerically stable algorithm is an important direction for future work.

## Broader Impacts

**Potential impacts to AI alignment** Dohmatob et al. [19] reported that repeated sampling from generative AIs may shift word frequency distributions toward lighter-tailed distributions. This may reduce linguistic diversity and lead to cultural homogenization by diminishing region-specific or

culturally unique expressions. Our Zipfian whitening and similar regularization methods could enhance output diversity, enriching the linguistic landscape.

**Potential negative societal impacts** The sentence similarity tasks used in our evaluation experiments are now considered core technologies for RAG (retrieval-augmented generation), which is essential when large language models leverage external resources. If chatbots generate responses tailored to user ideologies or preferred information sources, it may result in negative societal impacts, including political agitation.

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

# A Explanation fo the Zipfian whitened word vectors will have a zero mean and be in an isotropic position in terms of expectation

The second step of PCA whitening involves decorrelating the dimensions and then normalizing them, which is achieved by transforming the centered random vector $\overline{\boldsymbol{w}}$ as follows:

$$\widetilde{\boldsymbol{w}} := \mathbb{E}[\overline{\boldsymbol{w}}\,\overline{\boldsymbol{w}}^\top]^{-1/2}\overline{\boldsymbol{w}}. \tag{22}$$

Actually, $\mathbb{E}[\widetilde{\boldsymbol{w}}\widetilde{\boldsymbol{w}}^\top] = \boldsymbol{I}_d$ holds; $\widetilde{\boldsymbol{w}}$ satisfies Def. 2. Computationally, the estimation of $\mathbb{E}[\overline{\boldsymbol{w}}\,\overline{\boldsymbol{w}}^\top]^{-1/2}$ can be performed efficiently via singular value decomposition (SVD) of the centered "data" matrix $\overline{\boldsymbol{W}} := [\overline{\boldsymbol{w}}_1^\top, \ldots, \overline{\boldsymbol{w}}_n^\top]^\top$. Note again that, this standard method assumes that the frequency of each word (i.e., each row) is uniform, which presents the issues discussed in § 2. To account for word frequency, SVD should be performed on the matrix

$$\overline{\boldsymbol{W}}_p := \left[ \sqrt{p(w_1)}\overline{\boldsymbol{w}}_1^\top, \ldots, \sqrt{p(w_{|\mathcal{V}|})}\overline{\boldsymbol{w}}_{|\mathcal{V}|}^\top \right]^\top \in \mathbb{R}^{|\mathcal{V}| \times d}, \tag{23}$$

where each word frequency is multiplied by its *square root*. In fact, $\overline{\boldsymbol{W}}_p^\top \overline{\boldsymbol{W}}_p$ serves as an estimator for $\mathbb{E}_{\boldsymbol{w} \sim p}[\overline{\boldsymbol{w}}\,\overline{\boldsymbol{w}}^\top]$. This can be confirmed by comparing the $(j, k)$'th elements of each matrix (or matrix-valued random variable):

$$\mathbb{E}_{\boldsymbol{w} \sim p}[\overline{\boldsymbol{w}}\,\overline{\boldsymbol{w}}^\top][j, k] = \mathbb{E}_{\boldsymbol{w} \sim p}[\overline{\boldsymbol{w}}[j]\,\overline{\boldsymbol{w}}[k]], \tag{24}$$

$$(\overline{\boldsymbol{W}}_p^\top \overline{\boldsymbol{W}}_p)[j, k] = \sum_i p(w_i)\overline{\boldsymbol{w}}_i[j]\,\overline{\boldsymbol{w}}_i[k], \tag{25}$$

where $\boldsymbol{A}[j, k]$ denotes the $(j, k)$'th element of $\boldsymbol{A}$, and $\boldsymbol{v}[j]$ denotes the $j$'th element of $\boldsymbol{v}$. Finally, the estimator for the desired $\mathbb{E}[\overline{\boldsymbol{w}}\,\overline{\boldsymbol{w}}^\top]^{-1/2}$ can be computed as

$$\overline{\boldsymbol{W}}_p = \boldsymbol{U}\boldsymbol{\Sigma}\boldsymbol{V}^\top \quad \text{(via SVD)} \tag{26}$$

$$\widehat{\mathbb{E}}_{\boldsymbol{w} \sim p}[\overline{\boldsymbol{w}}\,\overline{\boldsymbol{w}}^\top]^{-1/2} = (\overline{\boldsymbol{W}}_p^\top \overline{\boldsymbol{W}}_p)^{-1/2} = ((\boldsymbol{U}\boldsymbol{\Sigma}\boldsymbol{V}^\top)^\top(\boldsymbol{U}\boldsymbol{\Sigma}\boldsymbol{V}^\top))^{-1/2} = (\boldsymbol{V}\boldsymbol{\Sigma}^2\boldsymbol{V}^\top)^{-1/2} = \boldsymbol{V}\boldsymbol{\Sigma}^{-1}. \tag{27}$$

Table 6 shows the correspondence between uniform (normal) whitening and Zipfian whitening. This may be useful for readers familiar with matrix notation.

# B Experimental settings

To ensure the reproducibility of the experiments conducted in this paper, we provide detailed configurations below. Additionally, the source code has been made publicly available at `https://github.com/cl-tohoku/zipfian-whitening`.

## B.1 Word embeddings

For static embeddings, we used the most standard ones, 300-dim **GloVe**[43] model trained on Common Crawl[8], 300-dim **word2vec**[37] model trained on Google News[9], and 300-dim **fastText**[11] subword/non-subword models trained on Common Crawl[10]. For the multilingual experiment, we used **fastText-ja** [11], a fastText model trained on Japanese Wikipedia and Common Crawl [11].

---

[8]`https://huggingface.co/sentence-transformers/average_word_embeddings_glove.6B.300d` from Sentence-Transformer's implementation [46].
[9]`https://drive.google.com/file/d/0B7XkCwpI5KDYNlNUTT1SS21pQmM/` from `https://code.google.com/archive/p/word2vec/`
[10]`https://fasttext.cc/docs/en/english-vectors.html`
[11]`https://fasttext.cc/docs/en/crawl-vectors.html`

Table 6: The correspondence between uniform (normal) whitening and Zipfian whitening.

| | Uniform whitening for general table data | Zipfian whitening for word embeddings | |
|---|---|---|---|
| data matrix, original vectors | $\boldsymbol{X} = \begin{bmatrix} \vdots \\ \boldsymbol{x}_i \\ \vdots \end{bmatrix}$ | $\boldsymbol{W} = \begin{bmatrix} \vdots \\ \boldsymbol{w}_i \\ \vdots \end{bmatrix}$ | $\in \mathbb{R}^{n \times d}$ |
| probability | $\boldsymbol{x} \sim$ Uniform dist. $p(\boldsymbol{x}_i) = \dfrac{1}{n}$ | $\boldsymbol{w} \sim$ Power-law dist. $p(\boldsymbol{w}_i) = p(w_i)$ (word freq.) | |
| mean vector | $\widehat{\boldsymbol{\mu}}_x = \sum_i \dfrac{1}{n} \boldsymbol{x}_i$ | $\widehat{\boldsymbol{\mu}}_w = \sum_i p(w_i) \boldsymbol{w}_i$ | $\in \mathbb{R}^d$ |
| centered matrix, centered vectors | $\overline{\boldsymbol{X}} = \begin{bmatrix} \vdots \\ \overline{\boldsymbol{x}}_i = \boldsymbol{x}_i - \widehat{\boldsymbol{\mu}}_x \\ \vdots \end{bmatrix}$ | $\overline{\boldsymbol{W}} = \begin{bmatrix} \vdots \\ \overline{\boldsymbol{w}}_i = \boldsymbol{w}_i - \widehat{\boldsymbol{\mu}}_w \\ \vdots \end{bmatrix}$ | $\in \mathbb{R}^{n \times d}$ |
| centered matrix to create cov. mat. | $\overline{\boldsymbol{X}}_p = \begin{bmatrix} \vdots \\ \sqrt{1/(n-1)}\, \overline{\boldsymbol{x}}_i \\ \vdots \end{bmatrix}$ | $\overline{\boldsymbol{W}}_p = \begin{bmatrix} \vdots \\ \sqrt{p(w_i)}\, \overline{\boldsymbol{w}}_i \\ \vdots \end{bmatrix}$ | $\in \mathbb{R}^{n \times d}$ |
| covariance matrix | $\widehat{\boldsymbol{S}} = \overline{\boldsymbol{X}}_p^\top \overline{\boldsymbol{X}}_p = \dfrac{\overline{\boldsymbol{X}}^\top \overline{\boldsymbol{X}}}{n-1}$ | $\widehat{\boldsymbol{S}} = \overline{\boldsymbol{W}}_p^\top \overline{\boldsymbol{W}}_p$ | $\in \mathbb{R}^{d \times d}$ |
| SVD of centered matrix = eigendecomposition of covariance matrix | $\overline{\boldsymbol{X}} = \boldsymbol{U}\boldsymbol{\Sigma}\boldsymbol{V}^\top$ $\boldsymbol{\Sigma} = \mathrm{diag}(\sigma_1, \ldots, \sigma_d)$ $\widehat{\boldsymbol{S}} = \dfrac{\overline{\boldsymbol{X}}^\top \overline{\boldsymbol{X}}}{n-1} = \boldsymbol{V}\boldsymbol{\Lambda}\boldsymbol{V}^\top$ $\boldsymbol{\Lambda} = \mathrm{diag}(\lambda_1, \ldots, \lambda_d)$ $:= \dfrac{\boldsymbol{\Sigma}^2}{n-1} = \mathrm{diag}\left(\dfrac{\sigma_1^2}{n-1}, \ldots\right)$ | $\overline{\boldsymbol{W}}_p = \boldsymbol{U}\boldsymbol{\Sigma}\boldsymbol{V}^\top$ $\boldsymbol{\Sigma} = \mathrm{diag}(\sigma_1, \ldots, \sigma_d)$ $\widehat{\boldsymbol{S}} = \overline{\boldsymbol{W}}_p^\top \overline{\boldsymbol{W}}_p = \boldsymbol{V}\boldsymbol{\Lambda}\boldsymbol{V}^\top$ $\boldsymbol{\Lambda} = \mathrm{diag}(\lambda_1, \ldots, \lambda_d)$ $:= \boldsymbol{\Sigma}^2 = \mathrm{diag}\left(\sigma_1^2, \ldots\right)$ | |
| whitened matrix whitened vector | $\boldsymbol{\Lambda}^{-1/2} := \mathrm{diag}\left(\dfrac{1}{\sqrt{\lambda_1}}, \ldots\right)$ $= \mathrm{diag}\left(\sqrt{\dfrac{n-1}{\sigma_1^2}}, \ldots\right)$ $\widetilde{\boldsymbol{X}} = \overline{\boldsymbol{X}}\boldsymbol{V}\boldsymbol{\Lambda}^{-1/2}$ $= \begin{bmatrix} \vdots \\ \widetilde{\boldsymbol{x}}_i = \overline{\boldsymbol{x}}_i \boldsymbol{V}\boldsymbol{\Lambda}^{-1/2} \\ \vdots \end{bmatrix}$ | $\boldsymbol{\Lambda}^{-1/2} := \mathrm{diag}\left(\dfrac{1}{\sqrt{\lambda_1}}, \ldots\right)$ $= \mathrm{diag}\left(\dfrac{1}{\sigma_1}, \ldots\right)$ $\widetilde{\boldsymbol{W}} = \overline{\boldsymbol{W}}\boldsymbol{V}\boldsymbol{\Lambda}^{-1/2}$ $= \begin{bmatrix} \vdots \\ \widetilde{\boldsymbol{w}}_i = \overline{\boldsymbol{w}}_i \boldsymbol{V}\boldsymbol{\Lambda}^{-1/2} \\ \vdots \end{bmatrix}$ | |

For dynamic embeddings, we used three most standard masked language models, **BERT**[18][12], **RoBERTa**[33][13], and **DeBERTa** [25][14]. All three models are base size. To aggregate the dynamic word embeddings to create sentence embeddings, we follow the first-last average pooling from the prior work [22]. In this setting, we first average the hidden states of first and last dynamic layer of

---

[12] https://huggingface.co/google-bert/bert-base-uncased
[13] https://huggingface.co/FacebookAI/roberta-base
[14] https://huggingface.co/microsoft/deberta-base

the model to get the averaged token embeddings[15], then average the token embeddings to get final sentence embeddings[16].

## B.2 Empirical word frequency and vocabulary

As the empirical word probability $p(w)$ of English words, we used the **enwiki** dataset preprocessed by Arora et al. [7][17]. For the Japanese word probability, we used Japanese Wikipedia word frequency from Wiktionary, denoted as **jawiki** [18]. Furthermore, we also used the frequency of words in the evaluation data itself (**test set probability**). The word frequency in the test set is implicitly utilized in [7]'s sentence embedding method and is also a natural approach in the context of "covariate shift" [52].

As vocabulary $\mathcal{V}$, we used the overlapping entries between the word frequency list and the pre-trained word embedding model's vocabulary across all settings, including baseline methods.

## B.3 Baseline methods

As baselines for post-processing of word vectors, we used **ABTT** (all-but-the-top) [39], which established the trend of post-processing word vectors; and the strong baseline method by [7], the combination of **SIF** (smoothed inverse frequency) weighting and **CCR** (common component removal)[19]. We followed the hyperparameter choices of the original papers, with the dimensionality reduction parameter for ABTT set to $D := 3$, and the weighting parameter for SIF set to $a := 10^{-3}$.

## B.4 Extrinsic tasks

As downstream tasks, we used the most commonly utilized ones in the community, **STS12-16** [1, 2, 3, 4, 5], **STS-B** [15] and **SICK-R** [35]. For the multilingual experiments, we used **JSTS** (Japanese version of the STS) from JGLUE benchmark [29]. They are *sentence*-level similarity tasks and are standard for empirically evaluating the performance of *word* vectors[20][21]. These datasets consist of pairs of sentences and their semantic similarity rated by annotators. We first tokenized the dataset

---

[15]Note that, we apply centering/whitening operations to such *token* embeddings, not to the final *sentence* embeddings, in order to match the setting in the theoretical analysis and the static word embedding experiments.

[16] Though we followed the experimental setting from the prior work [22], there is a slight discrepancy in the experimental results of the baseline setting. We found that this was due to prior work inadvertently taking the average of the hidden states of the zero-th layer (i.e., *static* word embedding layer) and the final dynamic layer. See the discussion at https://github.com/princeton-nlp/SimCSE/issues/285 for more details.

[17]https://github.com/PrincetonML/SIF/raw/master/auxiliary_data/enwiki_vocab_min200.txt

[18]https://en.wiktionary.org/wiki/Wiktionary:Frequency_lists/Japanese2015_10000 and https://en.wiktionary.org/wiki/Wiktionary:Frequency_lists/Japanese2015_10001-20000

[19]CCR is a process applied to *sentence* vectors, but due to its linearity, it can be adapted to *word* vectors. For more details, please refer to Yokoi et al. [60].

[20]For those outside the field of NLP research or practice, the question, "Why not run word-level evaluation metrics?" is a natural and valid one. Our language has a property known as compositionality, allowing infinite semantic content to be conveyed through a finite vocabulary as building blocks. This principle underlies models like word2vec [37], BERT [18], and the GPT series [12], where the fundamental unit of representation is the word; these models are then applied to solve tasks with larger components, such as sentences. Our research adheres to this foundational principle of NLP. Also, existing word-level similarity datasets have significant issues that make them less suitable for our work (see Bakarov [8, Section 4.1.1]). Given that whitening reflects word information content in vector norms, classic tasks like keyword extraction—which selects words with high information content—could be good candidates; results from a prior study using methods similar to ours would also be informative [42, Section 7.1].

[21]Setting aside the criticisms from previous studies for now, we conducted an evaluation using the two most well-known lexical similarity datasets. Table 7 shows the results. We found that the process of raw → Zipfian centering → Zipfian whitening consistently improves lexical properties. However, note that the finding "*direction: uniform whitening > direction: Zipfian whitening*" contradicts the experimental results in Appendix G, which showed "*direction: uniform whitening*, norm: Zipfian whitening < *direction: Zipfian whitening*, norm: Zipfian whitening." Here, lexical similarity tasks rely solely on vector direction and do not reference vector norms, as only the cosine similarity between word vectors is used to predict similarity. This discrepancy likely arises because these datasets are not representative of natural language, as discussed in Bakarov [8, Section 4.1.1]. For example, the widely used dataset WordSim353 [31] includes only about 200 subjective ratings on

by NLTK [9] with some post-processing following [20][22], then lowercased all tokens. The typical experimental protocol we followed is to sum the word vectors to form a "sentence vector" and then check if the angles (cosine similarity) between them correlate well with the gold scores. We reported Spearman's rank correlation between the predictions (cosine scores) and human-annotated gold scores[23].

## B.5 Computational resources for experiments

We conducted all experiments using a single NVIDIA RTX 6000 Ada GPU with 48GB VRAM. Each STS task required 10 seconds per model and whitening method, totaling approximately 10 minutes for the entire experiment, excluding the embedding loading time to the GPU.

For the calculation of the symmetry scores, each setting took one minute, resulting in a total of 5 minutes, again excluding the embedding loading time and the average cosine similarity (Ave. Cos.) setting. The Ave. Cos. score computation took 10 minutes per model, totaling 20 minutes for the two models.

## C Experimental results on all benchmark datasets to evaluate the effects of Zipfian whitening

In § 3.2, we evaluated the empirical performance of Zipfian whitening on the STS-B dataset. In this section, we present experimental results using more comprehensive datasets. Detailed experimental settings can be found in Appendix B. Table 8, Table 9 and Table 10 show the results. Across all datasets, the method incorporating a Zipfian prior consistently outperforms the method employing a uniform prior.

## D Proof of Prop. 1

*Proof.* We will show the following.

$$\mathbb{E}[(\boldsymbol{v} - \mathbb{E}[\boldsymbol{v}])(\boldsymbol{v} - \mathbb{E}[\boldsymbol{v}])^{\top}] \propto \boldsymbol{I}_d \tag{28}$$

$$\stackrel{\text{①}}{\Longleftrightarrow} \lambda_1 = \lambda_2 = \cdots = \lambda_d \tag{29}$$

$$\stackrel{\text{②}}{\Longleftrightarrow} \mathrm{Sym}_2(\boldsymbol{v}) = \frac{1}{\log d} H\left(\frac{\lambda_1}{\sum_j \lambda_j}, \ldots, \frac{\lambda_d}{\sum_j \lambda_j}\right) = 1 \tag{30}$$

( $\stackrel{\text{①}}{\Longrightarrow}$ ) When $\mathbb{E}[(\boldsymbol{v} - \mathbb{E}[\boldsymbol{v}])(\boldsymbol{v} - \mathbb{E}[\boldsymbol{v}])^{\top}] = k\boldsymbol{I}_d$ holds, its eigenvalues are $\{k, \ldots, k\}$.

( $\stackrel{\text{①}}{\Longleftarrow}$ ) Since $\mathbb{E}[(\boldsymbol{v} - \mathbb{E}[\boldsymbol{v}])(\boldsymbol{v} - \mathbb{E}[\boldsymbol{v}])^{\top}]$ is symmetric positive definite, it can be represented as $\boldsymbol{U}\boldsymbol{\Lambda}\boldsymbol{U}^{\top}$ using a diagonal matrix with eigenvalues $\boldsymbol{\Lambda} = \mathrm{diag}(\lambda_1, \ldots, \lambda_d) \in \mathbb{R}^{d \times d}$ and an orthogonal

---

common nouns, such as (tiger, cat, 7.35) or (king, cabbage, 0.23), which may or may not co-occur in the same document.

Table 7: Each cell shows the correlation coefficients $\times 100$ between the cosine similarity of (corrected) GloVe embeddings and the human-annotated gold score on lexical similarity tasks.

| | WordSim353 [31] | | | MEN [13] | |
|---|---|---|---|---|---|
| GloVe | 78.70 | | GloVe | 80.49 | |
| | Uniform | Zipfian | | Uniform | Zipfian |
| + Centering | 75.39 | **79.66** | + Centering | 78.07 | **80.55** |
| + Whitening | **82.31** | 80.90 | + Whitening | **84.35** | 83.97 |

[22]https://github.com/kawine/usif/blob/71ffef5b6d7295c36354136bfc6728a10bd25d32/usif.py#L113-L126

[23]We used the MTEB [40] implementation: https://github.com/embeddings-benchmark/mteb, for the evaluation of the static word embeddings in Table 2, Table 8, and Table 9. For the evaluation of the dynamic word embeddings in Table 5 and Table 12, we used the implementation in SimCSE paper [22]: https://github.com/princeton-nlp/SimCSE, to match the experimental setting.

matrix $\boldsymbol{U}$. Now we have $\lambda_1 = \lambda_2 = \ldots \lambda_d =: k$, then $\mathbb{E}[(\boldsymbol{v} - \mathbb{E}[\boldsymbol{v}])(\boldsymbol{v} - \mathbb{E}[\boldsymbol{v}])^\top] = \boldsymbol{U}\boldsymbol{\Lambda}\boldsymbol{U}^\top = \boldsymbol{U}k\boldsymbol{I}_d\boldsymbol{U}^\top = k\boldsymbol{I}_d$.

( $\overset{②}{\Longleftrightarrow}$ ) The Shannon entropy $H(p)$ of a random variable $p$ taking $d$ possible values attains its maximum value $\log d$ if and only if $p$ follows uniform distribution. $\qquad\square$

# E  Pseudocode for the evaluation metrics of symmetry

See Algorithm 2 to measure the degree of symmetry of word embeddings.

---

**Algorithm 2** Measure the degree of symmetry of word embeddings

---

**Input:** Word embeddings $\{\boldsymbol{w}_i\}$, word frequency $p \colon \mathcal{V} \to [0,1]$.

**Output:** Degree of centrality (the 1st moment) $\widehat{\mathrm{Sym}}_1(\{\boldsymbol{w}_i\}, p)$ and isotropy (the 2nd moment) $\widehat{\mathrm{Sym}}_2(\{\boldsymbol{w}_i\}, p)$.

  *Measure the degree of centrality (the 1st moment of symmetry):*
1:  $\widehat{\boldsymbol{\mu}} \leftarrow \sum_{w_i \in \mathcal{V}} p(w_i)\boldsymbol{w}_i \in \mathbb{R}^d$
2:  $\ell \leftarrow \sum_{w_i \in \mathcal{V}} p(w_i)\|\boldsymbol{w}_i\| \in \mathbb{R}$
3:  $\widehat{\mathrm{Sym}}_1(\{\boldsymbol{w}_i\}, p) \leftarrow 1 - \|\widehat{\boldsymbol{\mu}}\|/\ell$
  *Measure the degree of isotropy (the 2nd moment of symmetry):*
4:  $\boldsymbol{W} \leftarrow \left[ \sqrt{p(w_1)}(\boldsymbol{w}_1 - \widehat{\boldsymbol{\mu}})^\top, \ldots, \sqrt{p(w_{|\mathcal{V}|})}(\boldsymbol{w}_{|\mathcal{V}|} - \widehat{\boldsymbol{\mu}})^\top \right]^\top$
5:  $\boldsymbol{U}\boldsymbol{\Sigma}\boldsymbol{V}^\top \leftarrow \mathrm{SVD}(\mathcal{W})$
    $\triangleright\ \boldsymbol{\Sigma} = \mathrm{diag}(\sigma_1, \ldots, \sigma_d) \in \mathbb{R}^{d \times d}$ consists the singular values of $\boldsymbol{W}$.
6:  $(\lambda_1, \ldots, \lambda_d) \leftarrow (\sigma_1^2, \ldots, \sigma_d^2)$
7:  $\widehat{\mathrm{Sym}}_2(\{\boldsymbol{w}_i\}, p) \leftarrow -\dfrac{1}{\log d} \sum_i \dfrac{\lambda_i}{\sum_i \lambda_i} \log \dfrac{\lambda_i}{\sum_i \lambda_i}$

---

# F  Proof of Thm. 1

*Proof.* By the assumption, the word and context vectors for the same word $t \in \mathcal{V}$, $\boldsymbol{w}(t)$ and $\boldsymbol{c}(t)$, are obtained through the linear embedding layer with weight tying, namely, $\boldsymbol{w}(t) = \boldsymbol{U}\mathbf{1}_t$ and $\boldsymbol{c}(t) = \boldsymbol{U}\mathbf{1}_t$, where $\boldsymbol{U} \in \mathbb{R}^{d \times |\mathcal{V}|}$ is the embedding matrix and $\mathbf{1}_t \in \mathbb{R}^{|\mathcal{V}|}$ is the one-hot vector indicating the token $t$. To derive the KL divergence for the model $p(c \mid w)$, we need to begin with the generative model $p(w \mid c)$ (7) and confirm that $p(c \mid w)$ belongs to an exponential family.

$$p(c \mid w) = \frac{p(w \mid c)p(c)}{p(w)} = \frac{\exp(\langle \boldsymbol{w}, \boldsymbol{c} \rangle)}{Z_{\text{ⓩ}}(c)}p(c) = \frac{p(c)\exp(\langle \boldsymbol{U}^\top \boldsymbol{w}, \mathbf{1}_c \rangle)}{Z_{\text{ⓩ}}},$$

where we used the constancy of the partition function ($\equiv Z_{\text{ⓩ}}$) from (9) at the last identity. Hence, $p(c \mid w)$ is an exponential family parametrized by $\boldsymbol{U}^\top \boldsymbol{w}(:= \boldsymbol{\theta} \in \mathbb{R}^{|\mathcal{V}|})$, and its log-partition function is given as follows:

$$\psi_{c|w}(\boldsymbol{\theta}) = \log\left( \sum_{c \in \mathcal{V}} p(c)\exp(\langle \boldsymbol{\theta}, \mathbf{1}_c \rangle) \right).$$

The second-order expansion of the KL divergence can be derived based on the second moment of $p(c \mid w)$, which is given by the Hessian of $\psi_{c|w}$ in the case of exponential families. First, let us derive

the first moment.

$$\frac{\partial \psi_{c|w}}{\partial \boldsymbol{\theta}} = \frac{\sum_c \frac{p(c)}{Z_{\textcircled{z}}} \exp(\langle \boldsymbol{\theta}, \mathbf{1}_c \rangle) \mathbf{1}_c}{\sum_{c'} \frac{p(c')}{Z_{\textcircled{z}}} \exp(\langle \boldsymbol{\theta}, \mathbf{1}_{c'} \rangle)} = \sum_{c \in \mathcal{V}} \frac{p(c) \exp(\langle \boldsymbol{\theta}, \mathbf{1}_c \rangle)}{Z_{\textcircled{z}}} \mathbf{1}_c = \sum_{c \in \mathcal{V}} p(c \mid w) \mathbf{1}_c = \begin{bmatrix} \vdots \\ p(c \mid w) \\ \vdots \end{bmatrix}.$$

Then, the second moment is derived.

$$\frac{\partial \psi_{c|w}}{\partial \boldsymbol{\theta} \partial \boldsymbol{\theta}^\top} = \frac{1}{Z_{\textcircled{z}}} \sum_{c \in \mathcal{V}} p(c) \left\{ \frac{\partial}{\partial \boldsymbol{\theta}} \exp(\langle \boldsymbol{\theta}, \mathbf{1}_c \rangle) \right\} \mathbf{1}_c^\top = \frac{1}{Z_{\textcircled{z}}} \sum_{c \in \mathcal{V}} p(c) \exp(\langle \boldsymbol{\theta}, \mathbf{1}_c \rangle) \mathbf{1}_c \mathbf{1}_c^\top$$

$$= \sum_{c \in \mathcal{V}} p(c \mid w) \mathbf{1}_c \mathbf{1}_c^\top = \mathrm{diag}[\ldots \, p(c \mid w) \, \ldots],$$

which is the $|\mathcal{V}| \times |\mathcal{V}|$ diagonal matrix with $p(c \mid w)$ being the $(c, c)$-th diagonal entry. Now, we are ready to derive the KL divergence. For two tokens $w, w' \in |\mathcal{V}|$, if we write $\boldsymbol{\theta} := \boldsymbol{U} \mathbf{1}_w$ and $\boldsymbol{\theta}' := \boldsymbol{U} \mathbf{1}_{w'}$, the KL divergence of the exponential family can be expanded as the following quadratic form in their parameters $\boldsymbol{\theta}$ and $\boldsymbol{\theta}'$:

$$2\mathrm{KL}(p(\cdot \mid w') \| p(\cdot \mid w)) \approx (\boldsymbol{\theta}' - \boldsymbol{\theta})^\top \left( \frac{\partial \psi_{c|w}}{\partial \boldsymbol{\theta} \partial \boldsymbol{\theta}^\top} \right) (\boldsymbol{\theta}' - \boldsymbol{\theta})$$

$$= (\boldsymbol{w}' - \boldsymbol{w})^\top \boldsymbol{U} \left( \frac{\partial \psi_{c|w}}{\partial \boldsymbol{\theta} \partial \boldsymbol{\theta}^\top} \right) \boldsymbol{U}^\top (\boldsymbol{w}' - \boldsymbol{w})$$

$$= (\boldsymbol{w}' - \boldsymbol{w})^\top \left\{ \sum_{c \in |\mathcal{V}|} p(c \mid w)(\boldsymbol{U} \mathbf{1}_c)(\boldsymbol{U} \mathbf{1}_c)^\top \right\} (\boldsymbol{w}' - \boldsymbol{w})$$

$$= (\boldsymbol{w}' - \boldsymbol{w})^\top \left\{ \sum_{c \in \mathcal{V}} p(c \mid w) \boldsymbol{c} \boldsymbol{c}^\top \right\} (\boldsymbol{w}' - \boldsymbol{w})$$

$$= (\boldsymbol{w}' - \boldsymbol{w})^\top \boldsymbol{G}(w)(\boldsymbol{w}' - \boldsymbol{w})$$

$$= \| \boldsymbol{w}' - \boldsymbol{w} \|_{\boldsymbol{G}(w)}^2.$$

We can consider a word $w_0$ such that $p(\cdot) = p(\cdot \mid w_0)$, that is, an uninformative word $w_0$ whose presence does not change the marginal distribution at all. Noting from Equation (22) of Oyama et al. [42] that $\overline{\boldsymbol{w}} := \sum_{w \in \mathcal{V}} p(w) \boldsymbol{w} \approx \boldsymbol{w}_0$, we have

$$\mathrm{KL}(p(\cdot) \| p(\cdot \mid w)) = \mathrm{KL}(p(\cdot \mid w_0) \| p(\cdot \mid w)) \tag{31}$$

$$= \| \boldsymbol{w}_0 - \boldsymbol{w} \|_{\boldsymbol{G}(w)}^2 \tag{32}$$

$$\approx \| \overline{\boldsymbol{w}} - \boldsymbol{w} \|_{\boldsymbol{G}(w)}^2 \tag{33}$$

$$\overset{\text{Assump.}}{=} \| \boldsymbol{w} \|_{\boldsymbol{G}(w)}^2. \tag{34}$$

$\square$

# G  Ablation Study: Effects of Zipfian Whitening on Norm and Direction

In the main text, we demonstrated that applying Zipfian whitening to word vectors consistently improves downstream task performance (§ 3.2), and the theoretical and empirical evidence that Zipfian whitening has a beneficial impact on word vector norms (§ 4.2.1). Building on these findings, some readers may be interested in how the Zipfian whitening process influences the lengths (*norms*) and *directions* of word vectors individually. In this section, we present an ablation study to analyze the separate effects of Zipfian whitening on the lengths and directions of word vectors.

## Settings

The experimental settings here are identical to those used in the experiments presented so far—namely, those for Table 2 (excerpted results, shown in § 3.2) and Table 8 (comprehensive results, shown in Appendix C). For more detailed settings, refer to Appendix B.

**Results**

Table 11 presents the results of the ablation study. Here, "w/ Z.F. norm" refers to the process of "correcting word vectors under the uniform prior and then replacing only the norm with that obtained from Zipfian whitening (Z.F.)." Similarly, "w/ Z.F. direction" refers to "correcting them under the uniform prior and then replacing only the direction with that obtained from Z.F." These experiments allow us to isolate and examine the individual effects of Zipfian whitening on vector norms and directions. Here are some interesting findings from the experimental results.

First, replacing either the norm or the direction with its Zipfian whitening (Z.F.) version post hoc generally improves downstream task performance (see the "Ave." columns). This indicates that both the norm, as detailed in § 4.2.1, and the direction, which governs all degrees of freedom except for length, transition to a better-conditioned state through Zipfian whitening.

Next, regarding the relative contributions of norm and direction, we observe that the norm—despite being one-dimensional—has a greater impact on task performance than the direction. The reasons behind this, as well as theoretical insights related to isotropy and vector direction (or the mathematical understanding of cosine similarity as a conventional metric), remain open questions for future research.

Finally, it is worth noting that applying pure Zipfian whitening—leveraging its full effect on both norm and direction—yields the best average performance. There is no apparent trade-off here, suggesting that, from a practical standpoint, simply applying Zipfian whitening is the most effective approach.

# H  Formal Explanation of "Uniform whitening of token embeddings $\approx$ Zipfian whitening of type embeddings"

In this section, we provide a more formal explanation of "Uniform whitening of token embeddings $\approx$ Zipfian whitening of type embeddings," as described in § 5.1. For intuitive explanations and related discussions, please refer to § 5.1.

## H.1  Uniform whitening of token embeddings $\approx$ Zipfian whitening of type embeddings

Assume that, when the type word of a token $t$ is $w$, the token embedding $\boldsymbol{t}$ aligns with the shared type embedding $\boldsymbol{w}$.

**Assumption 1.** *If* $\mathrm{type}(t) = w$, *then* $\boldsymbol{t} = \boldsymbol{w}$.

Note that this is a rough approximation, as token embeddings are dynamic and vary with context. Under this assumption, the unweighted mean of token embeddings $\widehat{\mathbb{E}}_{\textcircled{u}}[\boldsymbol{t}]$ obtained from a dataset $\mathcal{D}$ is asymptotically equivalent to a word-frequency-weighted (Zipfian) average of type embeddings $\mathbb{E}_{\textcircled{z}}[\boldsymbol{w}]$, as $|\mathcal{D}| \to \infty$:

$$\widehat{\mathbb{E}}_{\textcircled{u}}[\boldsymbol{t}] := \frac{1}{|\mathcal{D}|} \sum_{t \in \mathcal{D}} \boldsymbol{t} \stackrel{\mathrm{Assump.\ 1}}{\approx} \frac{1}{|\mathcal{D}|} \sum_{w \in \mathcal{V}} c_{\mathcal{D}}(w)\boldsymbol{w} \xrightarrow{|\mathcal{D}|\to\infty} \sum_{w \in \mathcal{V}} p(w)\boldsymbol{w} =: \mathbb{E}_{\textcircled{z}}[\boldsymbol{w}], \qquad (35)$$

where $c_{\mathcal{D}}$ denotes the count of type $w$ in $\mathcal{D}$: $c_{\mathcal{D}}(w) := \#\{t \in \mathcal{D} : \mathrm{type}(t) = w\}$.

## H.2  Pseudo-uniform whitening of token embeddings $\approx$ uniform whitening of type embeddings

To establish a baseline for centering/whitening token embeddings under uniform prior, we can apply a coefficient ${1}/{|\mathcal{V}_{\mathcal{D}}|} \cdot {1}/{c_{\mathcal{D}}(\mathrm{type}(t))}$ to each token embedding $t$, for removing type frequencies that are implicitly referenced. Here, $\mathcal{V}_{\mathcal{D}}$ denotes the vocabulary contained in $\mathcal{D}$: $\mathcal{V}_{\mathcal{D}} := \{w \in \mathcal{V} : \exists t \in \mathcal{D}, \mathrm{type}(t) = w\}$. The "pseudo-uniform" average $\widehat{\mathbb{E}}_{\textcircled{\tilde{u}}}[\boldsymbol{t}]$ calculated in this way is asymptotically equivalent to the uniform average of type embeddings $\mathbb{E}_{\textcircled{u}}[\boldsymbol{w}]$, under the previous assumption

(Assump. 1) that ignores the dynamic nature of token embeddings:

$$\widehat{\mathbb{E}}_{\widehat{w}}[\boldsymbol{t}] := \sum_{t \in \mathcal{D}} \boxed{\frac{1}{|\mathcal{V}_{\mathcal{D}}|} \frac{1}{c_{\mathcal{D}}(\text{type}(t))}} \boldsymbol{t} \overset{\text{Assump. 1}}{\approx} \sum_{w \in \mathcal{V}_{\mathcal{D}}} \frac{1}{|\mathcal{V}_{\mathcal{D}}|} \frac{\cancel{c_{\mathcal{D}}(w)}}{\cancel{c_{\mathcal{D}}(w)}} \boldsymbol{w} \xrightarrow{|\mathcal{D}| \to \infty} \sum_{w \in \mathcal{V}} \frac{1}{|\mathcal{V}|} \boldsymbol{w} =: \mathbb{E}_{\widehat{w}}[\boldsymbol{w}].$$

(36)

# I  Experimental results on all benchmark datasets to evaluate the effects of uniform whitening on token embeddings

In § 5.1, we evaluated the empirical performance of *uniform* whitening of dynamic *token* embeddings on the STS-B dataset. In this section, we present experimental results using more comprehensive datasets. Table 12 shows the results. Across all datasets, the methods implicitly incorporating a Zipfian prior consistently outperforms the method employing a uniform prior.

Table 8: Full results of the empirical performance of Zipfian whitening. Each cell shows Spearman's $\rho \times 100$. As empirical word frequency $p(w)$, we used **enwiki**. Across all models and tasks, Zipfian whitening outperforms powerful baseline methods.

| Method | | STS12 | STS13 | STS14 | STS15 | STS16 | SICK-R | STS-B | Avg. |
|---|---|---|---|---|---|---|---|---|---|
| *GloVe* | | | | | | | | | |
| (raw) | | 56.46 | 50.41 | 51.13 | 58.60 | 49.03 | 57.01 | 46.17 | 52.69 |
| Uniform | Centering | 55.54 | 46.32 | 49.67 | 56.03 | 46.90 | 56.44 | 45.17 | 50.87 |
| | Whitening | 53.31 | 62.45 | 57.93 | 68.68 | 58.69 | 57.92 | 52.21 | 58.74 |
| Zipfian | Centering | 54.52 | 69.20 | 60.87 | 69.82 | 62.61 | 58.01 | 52.25 | 61.04 |
| | Whitening | 57.76 | **72.22** | **67.04** | **76.80** | **71.72** | **61.80** | **66.92** | **67.75** |
| ABTT | | 52.67 | 67.38 | 59.40 | 69.53 | 60.71 | 58.56 | 54.28 | 60.36 |
| SIF + CCR | | **60.23** | 68.78 | 62.39 | 67.26 | 61.85 | 56.91 | 58.70 | 62.30 |
| *Word2Vec* | | | | | | | | | |
| (raw) | | 58.57 | 68.64 | 63.65 | 71.73 | 61.79 | 61.77 | 56.98 | 63.30 |
| Uniform | Centering | 58.17 | 67.34 | 62.19 | 70.15 | 59.60 | 61.39 | 55.85 | 62.10 |
| | Whitening | 56.53 | 66.95 | 62.77 | 72.42 | 61.05 | 62.74 | 56.03 | 62.64 |
| Zipfian | Centering | 56.89 | 69.95 | 65.08 | 73.91 | 65.71 | 62.18 | 58.84 | 64.65 |
| | Whitening | 56.16 | 70.33 | **67.20** | **76.60** | **70.99** | **62.52** | **66.50** | **67.19** |
| ABTT | | 55.53 | 69.32 | 63.13 | 72.25 | 60.98 | 62.02 | 56.98 | 62.89 |
| SIF + CCR | | **60.05** | **73.26** | 66.87 | 74.32 | 67.64 | 59.22 | 63.04 | 66.34 |
| *fastText* | | | | | | | | | |
| (raw) | | 57.94 | 68.97 | 62.37 | 72.26 | 63.59 | 59.99 | 59.82 | 63.56 |
| Uniform | Centering | 59.73 | 55.02 | 55.16 | 64.22 | 53.39 | 58.85 | 52.46 | 56.98 |
| | Whitening | 52.47 | 59.01 | 53.90 | 65.33 | 52.61 | 58.34 | 48.60 | 55.75 |
| Zipfian | Centering | 58.30 | 71.69 | 64.57 | 74.10 | 67.59 | 60.75 | 59.40 | 65.20 |
| | Whitening | 58.86 | 73.85 | **68.43** | **78.07** | **74.00** | **62.85** | **69.55** | **69.37** |
| ABTT | | 58.35 | 69.09 | 60.82 | 71.99 | 60.76 | 60.34 | 57.02 | 62.62 |
| SIF + CCR | | **61.54** | **76.95** | 68.39 | 76.98 | 70.27 | 59.52 | 67.08 | 68.67 |
| *fastText-subword* | | | | | | | | | |
| (raw) | | 49.10 | 47.34 | 51.94 | 61.99 | 51.54 | 53.60 | 50.43 | 52.28 |
| Uniform | Centering | 49.21 | 43.13 | 49.89 | 62.03 | 49.70 | 54.56 | 46.91 | 50.78 |
| | Whitening | 45.12 | 41.00 | 47.30 | 62.08 | 48.85 | 54.80 | 43.55 | 48.96 |
| Zipfian | Centering | 48.68 | 55.03 | 54.07 | 60.23 | 58.41 | 54.64 | 50.38 | 54.49 |
| | Whitening | **61.22** | **60.68** | **63.18** | **73.59** | **69.87** | **59.82** | **68.20** | **65.22** |
| ABTT | | 49.64 | 41.79 | 48.81 | 60.84 | 47.57 | 55.09 | 44.23 | 49.71 |
| SIF + CCR | | 57.28 | 54.50 | 60.77 | 68.82 | 61.63 | 56.83 | 60.36 | 60.03 |

Table 9: Full results of the empirical performance of Zipfian whitening, test set frequency setting. Each cell shows Spearman's $\rho \times 100$. As empirical word frequency $p(w)$, we used **test set frequency**. Across all models and tasks, Zipfian whitening outperforms powerful baseline methods. Besides, the test set frequency setting consistently outperforms the enwiki setting in Table 8, demonstrating that the models benefit from using task-specific statistics in line with a covariate shift approach [52].

| Method | | STS12 | STS13 | STS14 | STS15 | STS16 | SICK-R | STS-B | Avg. |
|---|---|---|---|---|---|---|---|---|---|
| *GloVe* | | | | | | | | | |
| (raw) | | 57.71 | 50.29 | 50.61 | 58.38 | 48.76 | 56.76 | 46.22 | 52.67 |
| Uniform | Centering | 56.32 | 61.17 | 52.68 | 64.80 | 55.80 | 57.98 | 47.94 | 56.67 |
| | Whitening | 51.67 | 60.94 | 57.14 | 70.09 | 63.08 | 55.14 | 53.16 | 58.74 |
| Zipfian | Centering | 50.69 | 70.66 | 61.59 | 70.19 | 68.25 | 60.03 | 56.64 | 62.58 |
| | Whitening | **61.63** | **78.36** | **69.48** | **76.83** | **74.08** | **60.11** | **71.60** | **70.30** |
| ABTT | | 52.93 | 66.93 | 60.10 | 71.93 | 63.12 | 58.23 | 53.72 | 60.99 |
| *Word2Vec* | | | | | | | | | |
| (raw) | | 59.00 | 68.92 | 63.99 | 72.51 | 62.25 | 61.87 | 57.15 | 63.67 |
| Uniform | Centering | 57.88 | 70.34 | 64.24 | 74.71 | 65.57 | 62.47 | 58.09 | 64.76 |
| | Whitening | 58.45 | 69.42 | 65.46 | 76.43 | 67.78 | 62.87 | 60.85 | 65.89 |
| Zipfian | Centering | 55.02 | 71.47 | 65.81 | 74.36 | 69.52 | 62.92 | 61.02 | 65.73 |
| | Whitening | **59.37** | **76.92** | **69.48** | **76.42** | **73.56** | **60.07** | **70.42** | **69.46** |
| ABTT | | 56.33 | 70.42 | 64.71 | 74.74 | 65.19 | 62.55 | 58.21 | 64.59 |
| *fastText* | | | | | | | | | |
| (raw) | | 58.23 | 69.36 | 62.89 | 73.09 | 64.25 | 60.22 | 60.27 | 64.04 |
| Uniform | Centering | 60.60 | 69.51 | 61.09 | 73.92 | 64.49 | 61.14 | 57.42 | 64.02 |
| | Whitening | 55.56 | 63.51 | 57.73 | 70.68 | 62.40 | 57.93 | 54.65 | 60.35 |
| Zipfian | Centering | 55.92 | 73.36 | 65.72 | 74.12 | 72.18 | 62.30 | 62.95 | 66.65 |
| | Whitening | **62.20** | **79.35** | **71.03** | **77.95** | **76.28** | **60.66** | **73.56** | **71.58** |
| ABTT | | 59.13 | 71.00 | 63.30 | 74.80 | 65.96 | 61.69 | 58.23 | 64.87 |
| *fastText-subword* | | | | | | | | | |
| (raw) | | 51.37 | 51.49 | 54.57 | 62.75 | 52.97 | 53.53 | 52.41 | 54.16 |
| Uniform | Centering | 51.31 | 44.80 | 49.66 | 62.27 | 47.43 | 54.86 | 43.12 | 50.49 |
| | Whitening | 51.52 | 49.33 | 53.51 | 68.28 | 58.34 | 56.94 | 51.69 | 55.66 |
| Zipfian | Centering | 43.15 | 53.40 | 53.67 | 63.05 | 59.09 | 56.57 | 47.16 | 53.73 |
| | Whitening | **60.87** | **72.21** | **67.79** | **75.86** | **73.88** | **60.52** | **70.99** | **68.87** |
| ABTT | | 49.06 | 45.16 | 49.57 | 62.14 | 50.75 | 55.49 | 44.53 | 50.96 |

Table 10: Evaluation results using Japanese fastText. Each cell shows the JSTS [29] score $\times 100$. Even in the multilingual setting, Zipfian whitening outperforms powerful baseline methods.

(a) With **jawiki** as $p(w)$

| fastText-ja | | 55.81 |
|---|---|---|
| | Uniform | Zipfian |
| + Centering | 56.05 | **57.55** |
| + Whitening | 55.53 | **65.56** |
| + ABTT | | 57.14 |
| + SIF + CCR | | 61.03 |

(b) With **test set frequency** as $p(w)$

| fastText-ja | | 59.94 |
|---|---|---|
| | Uniform | Zipfian |
| + Centering | 59.89 | **63.05** |
| + Whitening | 61.75 | **69.86** |
| + ABTT | | 63.02 |

Table 11: Ablation study on the effects of Zipfian whitening on norm and direction. Here, "w/ Z.F. norm" refers to the process of "correcting word vectors under the uniform prior and then replacing only the norm with that obtained from Zipfian whitening (Z.F.)." Similarly, "w/ Z.F. direction" refers to "correcting them under the uniform prior and then replacing only the direction with that obtained from Z.F." Each cell shows Spearman's $\rho \times 100$. Except for the "w/ Z.F. norm" and "w/ Z.F. direction" settings, the values are simply copied from Table 8. The results indicate that both the norm and the direction transition to a better-conditioned state through Zipfian whitening.

| Method | | | STS12 | STS13 | STS14 | STS15 | STS16 | SICK-R | STS-B | Avg. |
|---|---|---|---|---|---|---|---|---|---|---|
| *GloVe* | | | | | | | | | | |
| (raw) | | | 56.46 | 50.41 | 51.13 | 58.60 | 49.03 | 57.01 | 46.17 | 52.69 |
| | | w/ Z.F. norm | **62.08** | 66.62 | 59.34 | 65.93 | 58.57 | 57.52 | 55.43 | 60.78 |
| | | w/ Z.F. direction | 49.66 | 64.15 | 61.73 | 71.89 | 66.37 | 59.52 | 57.20 | 61.50 |
| Uniform | Centering | | 55.54 | 46.32 | 49.67 | 56.03 | 46.90 | 56.44 | 45.17 | 50.87 |
| | Whitening | | 53.31 | 62.45 | 57.93 | 68.68 | 58.69 | 57.92 | 52.21 | 58.74 |
| Uniform | Centering | w/ Z.F. norm | 61.41 | 63.94 | 58.33 | 63.88 | 56.90 | 56.39 | 54.62 | 59.35 |
| | Whitening | w/ Z.F. norm | 60.20 | **73.63** | 64.98 | 74.21 | 66.67 | 60.53 | 61.55 | 65.97 |
| Uniform | Centering | w/ Z.F. direction | 47.12 | 61.14 | 59.42 | 70.10 | 63.46 | 58.58 | 53.87 | 59.10 |
| | Whitening | w/ Z.F. direction | 49.57 | 65.00 | 62.17 | 72.24 | 67.83 | 58.22 | 57.73 | 61.82 |
| Zipfian | Centering | | 54.52 | 69.20 | 60.87 | 69.82 | 62.61 | 58.01 | 52.25 | 61.04 |
| | Whitening | | 57.76 | 72.22 | **67.04** | **76.80** | **71.72** | **61.80** | **66.92** | **67.75** |
| *Word2Vec* | | | | | | | | | | |
| (raw) | | | 58.57 | 68.64 | 63.65 | 71.73 | 61.79 | 61.77 | 56.98 | 63.30 |
| | | w/ Z.F. norm | 58.87 | **71.96** | 65.52 | 73.55 | 66.61 | 61.05 | 61.59 | 65.59 |
| | | w/ Z.F. direction | 55.45 | 68.07 | 66.49 | 75.98 | 68.46 | **63.41** | 63.25 | 65.87 |
| Uniform | Centering | | 58.17 | 67.34 | 62.19 | 70.15 | 59.60 | 61.39 | 55.85 | 62.10 |
| | Whitening | | 56.53 | 66.95 | 62.77 | 72.42 | 61.05 | 62.74 | 56.03 | 62.64 |
| Uniform | Centering | w/ Z.F. norm | **59.01** | 71.91 | 65.28 | 73.26 | 65.71 | 61.34 | 61.27 | 65.40 |
| | Whitening | w/ Z.F. norm | 58.49 | 71.53 | 66.22 | 75.62 | 67.82 | 61.95 | 62.87 | 66.35 |
| Uniform | Centering | w/ Z.F. direction | 54.94 | 67.87 | 66.18 | 75.62 | 68.27 | 63.11 | 62.83 | 65.55 |
| | Whitening | w/ Z.F. direction | 53.40 | 66.91 | 65.15 | 75.11 | 66.90 | 63.24 | 60.95 | 64.52 |
| Zipfian | Centering | | 56.89 | 69.95 | 65.08 | 73.91 | 65.71 | 62.18 | 58.84 | 64.65 |
| | Whitening | | 56.16 | 70.33 | **67.20** | **76.60** | **70.99** | 62.52 | **66.50** | **67.19** |
| *fastText* | | | | | | | | | | |
| (raw) | | | 57.94 | 68.97 | 62.37 | 72.26 | 63.59 | 59.99 | 59.82 | 63.56 |
| | | w/ Z.F. norm | 61.35 | 75.69 | 66.77 | 75.69 | 69.41 | 61.24 | 65.31 | 67.92 |
| | | w/ Z.F. direction | 55.12 | 70.20 | 66.23 | 75.92 | 71.95 | 61.57 | 65.35 | 66.62 |
| Uniform | Centering | | 59.73 | 55.02 | 55.16 | 64.22 | 53.39 | 58.85 | 52.46 | 56.98 |
| | Whitening | | 52.47 | 59.01 | 53.90 | 65.33 | 52.61 | 58.34 | 48.60 | 55.75 |
| Uniform | Centering | w/ Z.F. norm | **65.26** | 70.32 | 63.63 | 71.24 | 62.99 | 60.34 | 61.24 | 65.00 |
| | Whitening | w/ Z.F. norm | 62.48 | **76.02** | 66.73 | 76.55 | 68.65 | 61.56 | 64.31 | 68.04 |
| Uniform | Centering | w/ Z.F. direction | 52.86 | 68.40 | 64.60 | 74.39 | 70.23 | 60.92 | 62.70 | 64.87 |
| | Whitening | w/ Z.F. direction | 46.45 | 64.51 | 60.10 | 70.59 | 64.80 | 58.58 | 54.87 | 59.99 |
| Zipfian | Centering | | 58.30 | 71.69 | 64.57 | 74.10 | 67.59 | 60.75 | 59.40 | 65.20 |
| | Whitening | | 58.86 | 73.85 | **68.43** | **78.07** | **74.00** | **62.85** | **69.55** | **69.37** |
| *fastText-subword* | | | | | | | | | | |
| (raw) | | | 49.10 | 47.34 | 51.94 | 61.99 | 51.54 | 53.60 | 50.43 | 52.28 |
| | | w/ Z.F. norm | 58.03 | **60.35** | 60.76 | 67.99 | 61.74 | 56.35 | 61.47 | 60.96 |
| | | w/ Z.F. direction | 47.65 | 49.71 | 55.26 | 67.96 | 61.81 | 56.62 | 54.72 | 56.25 |
| Uniform | Centering | | 49.21 | 43.13 | 49.89 | 62.03 | 49.70 | 54.56 | 46.91 | 50.78 |
| | Whitening | | 45.12 | 41.00 | 47.30 | 62.08 | 48.85 | 54.80 | 43.55 | 48.96 |
| Uniform | Centering | w/ Z.F. norm | 60.29 | 58.99 | 61.11 | 70.23 | 65.36 | 57.78 | 63.06 | 62.40 |
| | Whitening | w/ Z.F. norm | 61.06 | **60.35** | 63.07 | **74.29** | 69.19 | 59.56 | 65.09 | 64.66 |
| Uniform | Centering | w/ Z.F. direction | 46.17 | 49.23 | 54.49 | 67.33 | 61.11 | 55.95 | 53.01 | 55.33 |
| | Whitening | w/ Z.F. direction | 43.20 | 46.68 | 52.15 | 65.46 | 59.01 | 54.07 | 49.26 | 52.83 |
| Zipfian | Centering | | 48.68 | 55.03 | 54.07 | 60.23 | 58.41 | 54.64 | 50.38 | 54.49 |
| | Whitening | | **61.22** | 60.68 | **63.18** | 73.59 | **69.87** | **59.82** | **68.20** | **65.22** |

Table 12: Full results of the whitening on dynamic embeddings. Each cell shows the STS score ×100. Token-level uniform centering/whitening ("Zipfian" settings), which corresponds to centering/whitening at the word type level under a Zipfian prior, consistently outperforms the "Uniform" setting across all STS tasks.

| Method | | STS12 | STS13 | STS14 | STS15 | STS16 | SICK-R | STS-B | Avg. |
|---|---|---|---|---|---|---|---|---|---|
| *BERT-base uncased* | | | | | | | | | |
| First-last avg. | | 45.09 | 64.30 | 54.56 | 70.52 | 67.87 | 59.05 | 63.75 | 60.73 |
| "Uniform" | Centering | 47.51 | 64.53 | 54.68 | 72.19 | 69.28 | 59.77 | 64.04 | 61.71 |
| | Whitening | 40.31 | 56.11 | 47.02 | 68.35 | 64.53 | 48.59 | 60.53 | 55.06 |
| "Zipfian" | Centering | 47.58 | 66.26 | 57.32 | 73.18 | 71.09 | 63.27 | 64.82 | 63.36 |
| | Whitening | **53.75** | **74.07** | **64.21** | **73.88** | **72.83** | **69.71** | **64.91** | **67.62** |
| *RoBERTa-base* | | | | | | | | | |
| First-last avg. | | 44.00 | 59.02 | 49.31 | 66.63 | 59.62 | 57.56 | 60.75 | 56.70 |
| "Uniform" | Centering | 46.07 | 55.50 | 46.27 | 66.06 | 60.06 | 51.33 | 60.34 | 55.09 |
| | Whitening | 37.67 | 54.64 | 47.71 | 66.31 | 62.85 | 50.13 | 61.31 | 54.37 |
| "Zipfian" | Centering | 44.97 | 61.19 | 53.73 | 69.57 | 67.88 | 58.60 | 61.30 | 59.61 |
| | Whitening | **52.80** | **73.39** | **64.18** | **72.64** | **72.02** | **71.07** | **65.69** | **67.40** |
| *DeBERTa-base* | | | | | | | | | |
| First-last avg. | | 45.03 | 61.94 | 52.39 | 68.90 | 64.83 | 56.54 | 61.66 | 58.76 |
| "Uniform" | Centering | 45.20 | 61.25 | 50.84 | 68.56 | 63.87 | 53.18 | 62.01 | 57.84 |
| | Whitening | 38.12 | 50.46 | 45.30 | 63.52 | 62.29 | 46.99 | 58.19 | 52.12 |
| "Zipfian" | Centering | 45.87 | 63.24 | 55.07 | 70.53 | 68.88 | 58.50 | 63.18 | 60.75 |
| | Whitening | **52.97** | **73.54** | **63.25** | **72.60** | **71.97** | **69.79** | **64.63** | **66.96** |

