# OpenReview forum: "Zipfian Whitening"
_NeurIPS.cc/2024/Conference — NeurIPS 2024 poster_

### Official Review · Reviewer_9YCm · 2024-07-09

**Soundness:** 3
**Presentation:** 4
**Contribution:** 3
**Rating:** 7
**Confidence:** 3

**Summary:**

This paper proposes new Zipfian whitening for static word embeddings inspired by Zipf's law. The main idea is to use empirical word frequencies as a prior rather than using a uniform prior. The authors show the superiority of their method compared to previous widely-used baselines. The paper also presents the metric for measuring symmetry in word embeddings.

**Strengths:**

- Novel approach for inducing symmetry into the embedding space inspired by Zipf's law
- Easy to follow paper; nicely written
- Empirical results on downstream tasks show the efficiency of the proposed Zipfian whitening
- Analysis from different prospectives is presented
- Paper draws an interesting connection to the prior research

**Weaknesses:**

- Limited evaluation on word-level static embeddings: one language studied, one dataset/vocabulary and its frequencies

**Questions:**

- In the paper, you propose a method that relies heavily on the empirical frequency. Thus, it would be interesting to look at how changing the empirical distribution will affect the downstream task performance
- Also, as far as I understand, you discard lots of infrequent tokens (frequency less than 200 according to the enwiki vocab), while in my opinion, the most interesting effect can be seen on low-frequency tokens. What is your opinion on this matter? And how should you deal with the OOV embeddings (i.e., missing in the frequency distribution)? It is especially crucial for low-resource languages.
- While it's a minor point, I'm curious why you haven't explored token-level embeddings, given that it's a de-facto standard in the field today. For instance, there are static bpe level embeddings available (fasttext). Incorporating these could significantly bolster your claims.

**Limitations:**

I would encourage authors to add the limitations section addressing their empirical analysis scope.

---

> ### Author Rebuttal · Authors · 2024-08-06
>
> Thank you for your positive review! We're delighted to hear it. We especially appreciate your constructive feedback on the various aspects of our experimental setup. Below, we provide our responses.
>
> ### 1. Embedding models
>
> > While it's a minor point, I'm curious why you haven't explored token-level embeddings, given that it's a de-facto standard in the field today. For instance, there are static bpe level embeddings available (fasttext). Incorporating these could significantly bolster your claims.
>
> Absolutely! Using fastText is indeed a very natural and convincing experimental setup. Thank you for the suggestion. We quickly conducted similar experiments to those in Table 1 using fastText embeddings trained with Common Crawl and STS-B dataset.
>
> |fastText|Uniform|Zipfian|
> |:-|:-:|:-:|
> |raw|60.46|60.46|
> |+ Centering|51.39|61.64|
> |+ Whitening|48.55|72.26|
>
> The stable effects of Zipfian whitening were confirmed. We will include the results applied across the entire paper in the camera-ready version.
>
> ### 2. Word frequency
>
> > Also, as far as I understand, you discard lots of infrequent tokens (frequency less than 200 according to the enwiki vocab), while in my opinion, the most interesting effect can be seen on low-frequency tokens. What is your opinion on this matter? And how should you deal with the OOV embeddings (i.e., missing in the frequency distribution)? It is especially crucial for low-resource languages.
>
> Thank you for your very interesting question, which touches on important issues, the heavy tail of word frequency and low-resource languages. We agree that OOV words are inevitable since word frequencies follow a long-tail distribution. We will include several considerations below in the revised version.
>
> - **Algorithm** — Since whitening is a global affine transformation on the entire word embedding space (Algorithm 1), our post-processing is possible/applicable even if the frequency of a target word is unknown or extremely low.
> - **Empirical Results** — Based on your comment, we found it interesting to conduct experiments specifically designed to create cases with intentional OOV words (words with unknown frequency). We conducted experiments using only the top 1,000 word frequencies (0.5%) from `enwiki_vocab_min200`, similar to Table 1 with GloVe and STS-B. Results were largely maintained, suggesting the method's robustness to some OOV words.
>
> ||Zipfian|
> |:-|:-:|
> |raw|43.65|
> |+ Centering|55.15|
> |+ Centering (OOV settings)|57.59|
> |+ Whitening|70.22|
> |+ Whitening (OOV settings)|71.54|
>
> - **Implications** — The mean vector used in post-processing ($\hat\mu$ in Algorithm 1) and the weighted data matrix ($W_p$ in Algorithm 1) are not significantly influenced by extremely low-frequency words with small $p(w)$. Instead, the process primarily *removes signals from high-frequency words*. This suggests that even for low-resource languages, as long as the "head" is well-observed, the process can be applied and might preserve signals from the "tail" side. we will experimentally test this hypothesis using multilingual vectors (e.g. fastText) and the multilingual evaluation dataset including low-resource languages (Ousidhoum et al. "SemRel2024: A Collection of Semantic Textual Relatedness Datasets for 13 Languages" 2024), and include these results in the camera-ready version.
>
> Additionally, to clarify, the experiments conducted on English text, as reported in the paper, were carried out in a setting where OOV occurrences were minimal. Specifically, `the enwiki_vocab_min200` contains 188,033 words. For comparison, one of the current largest models, LLaMA 3.1 by Meta, has a vocab size of 128,000 (h/t "The Llama 3 Herd of Models" arXiv 2407.21783; though this is subword tokenized, so it's not a perfect one-to-one comparison). For reference, the lowest-frequency words in `enwiki_vocab_min200` with a frequency of 200 include minor proper nouns such as "abbi", "aberto", "abgrallaspis", "abhilasha", and "acurate" (which is not "accurate"). We have provided more detailed data in our response to Reviewer VABi. Also, there were no OOV issues when solving STS and SICK-R tasks in our settings. We will include a detailed explanation of this in the manuscript.
>
> ---
>
> If there are any discrepancies between our proposed experimental settings and your expectations, we would appreciate your further comments. Once again, thank you for your wide-ranging suggestions to make our empirical results more convincing!

---

> > ### Comment · Reviewer_9YCm · 2024-08-12
> >
> > Thank you for clarification and providing additional results, I appreciate that. Now I am even more confident in my previous assessment. I will keep my score.

---

### Official Review · Reviewer_jUzu · 2024-07-11

**Soundness:** 3
**Presentation:** 3
**Contribution:** 3
**Rating:** 7
**Confidence:** 4

**Summary:**

This paper considers the problem of post-processing static word embedding spaces based on the observation that the distribution is spatially skewed based on the occurrence frequency of corresponding words. The authors propose Zipfian whitening, an approach to symmetrize the embedding space using the empirical frequency distribution to weight the influence of each embedding. The authors present empirical evidence for their proposed approach on standard datasets and metrics. Theoretical connections to exponential family distributions provide intuition for why Zipfian whitening is better than uniform whitening.

**Strengths:**

*  The potential downstream impact of this paper is well beyond the immediate superficial contributions. The paper presents limited (yet profound!) results on small, unrealistic, and impractical datasets, but these results and observations, along with the theoretical connections, could influence how we address the long-tail in many aspects of training, evaluation, safety, and alignment. The technical contributions might not be directly applicable to all of these problems, but making the community aware of such an approach could inspire many other researchers to solve problems of this sort.
* The paper is written in a refreshingly unorthodox fashion. It does not follow the standard template of machine learning papers making it much more enjoyable to read and more likely to have a bigger impact.
* The paper addresses all aspects of the problem. The authors not only show that their proposed approach works, but also present empirical and theoretical evidence for why it works.

**Weaknesses:**

* The empirical results are quite impractical. It would be better if there was empirical evidence on some sort of dynamic or causal embeddings.
* Section 4 seems to be a bit of a stretch. It is understandable that there is limited space, but it is a quite hand-wavy explanation. I'm not sure how much this particular section contributes to the paper.
* It is not immediately obvious how this work could be applied to causal and dynamic large language models?

**Questions:**

* Could you more explicitly explain how uniform whitening of contextual word embeddings is related to Zipfian whitening?
* How would the proposed approach be used to regularize large language models trained on next token prediction?
* What is $c$ in Section 3? Is it a class label?
* If a word is very infrequent, is there a concern that it's embedding is of poor quality? In this case, would we be concerned that its poor quality would skew the centroid of the space?
* Could this approach be useful for understanding large language model embeddings?
* How do the insights provided in this paper inform proposed approaches in safety and alignment? (connections to alignment are mentioned in the Broader Impacts, but not explained) Could this work have impact on aspects of mechanistic interpretability?

**Limitations:**

The authors have thoroughly addressed the limitations and potential societal impact of their work.

---

> ### Author Rebuttal · Authors · 2024-08-06
>
> Thank you for your positive evaluation of our paper, including its potential impact on future research! We also appreciate your many insightful and constructive questions. We'll do our best to provide honest answers below. If any remaining discrepancies in our understanding or points need clarification, we'd be grateful for your further input.
>
> ### 1. Correction of contextual/causal embeddings
>
> > The empirical results are quite impractical. It would be better if there was **empirical evidence on some sort of dynamic or causal embeddings**.
>
> > Could you more explicitly explain **how uniform whitening of contextual word embeddings is related to Zipfian whitening**?
>
> We agree that the ML/NLP community's focus is on contextual and causal language models, and acknowledge our Section 4 lacks detail.
>
> In the revision, we'll add a section discussing our research's relationship with these models, starting with the **type-token distinction**. The type-token distinction is a well-known concept in linguistics and related fields, where *type* represents a class and *token* represents an instance. For example, the phrase "perform natural language processing in a natural way" contains eight tokens and *seven* types; the instances "natural" appear twice, but as a word type, it is counted only once.
>
> This distinction clarifies the relation between uniform whitening of contextual word embeddings and Zipfian whitening. Addition-based sentence embeddings are obtained by summing up *token* embeddings in a sentence. Given that a sufficiently long sentence may represent the underlying Zipfian word frequency, the uniform sampling on these tokens is approximately nothing else but the Zipfian sampling on *types*. That's why we claim that the uniform centering/whitening of sentence embeddings corresponds to the Zipfian centering/whitening of type embeddings. Our contribution provides a new explanation for the empirical success of existing methods like uniform centering (*1: Chen et al. 2020) and whitening (*2: Huang et al. 2021) of token vectors. To strengthen our argument, we'll include an experiment applying a *pseudo*-uniform prior by multiplying token embeddings by $1/p(w)$. We'll compare this with existing methods (*1, *2) that implicitly use a Zipfian prior!
>
> ### 2. Connection with other aspects of contextual/causal LMs
>
> > How would the proposed approach **be used to regularize large language models trained on next token prediction**?
>
> A promising direction is the orthogonalization of embedding matrices. Previous studies added regularizers to increase the effective rank or impose orthogonality of word (un)embedding matrices (e.g. Wang et al. "Improving Neural Language Generation With Spectrum Control" 2020). However, they treat the word (un)embedding matrix as a standard data matrix, thus implicitly assuming a uniform prior. For future work, we’d like to try to adopt a Zipfian prior to make vectors in the (un)embedding matrix effectively isotropic, thereby maximizing expressive power while accounting for word frequency.
>
> > How do the insights provided in this paper inform proposed approaches in **safety and alignment**? (connections to alignment are mentioned in the Broader Impacts, but not explained)
>
> Dohmatob et al. "A Tale of Tails: Model Collapse as a Change of Scaling Laws" (2024) reported that repeated sampling from generative AIs may shift word frequency distributions towards light-tailed ones. This could reduce linguistic diversity and cause cultural homogenization by decreasing region-specific or culturally unique expressions. Our Zipfian whitening and similar regularization methods can be used to enhance output diversity, thereby enriching the resulting linguistic landscape.
>
> > Could this work have impact on aspects of **mechanistic interpretability**?
>
> Regarding the type-token distinction, embedding and unembedding matrices in causal LMs primarily retain word *type* information. Thus, improving the logit lens approach, which analyzes hidden vectors by projecting them onto the unembedding matrix, could contribute to the mech. interp. community; the softmax function used after projection implicitly assumes a uniform prior.
>
> ### 3. Other points of discussion
>
> > What is $c$ in Section 3? Is it a class label?
>
> Thank you for pointing this out! Our description was lacking. $c$ represents context, generalizing the information used to predict a word $w$ in static/masked/causal LMs. Specifically, it represents a co-occurring word, a cloze sentence, or a prefix of a sentence. In all cases, prediction involves calculating $\langle \boldsymbol w, \boldsymbol c\rangle$. Furthermore, for a rigorous discussion, particularly when addressing Thm. 1 for example, we should restrict $c$ to co-occurring words. We will revise the manuscript to clarify this point.
>
> > If a word is very infrequent, is there a concern that it's embedding is of poor quality? In this case, would we be concerned that its poor quality would skew the centroid of the space?
>
> We appreciate your insightful comment. Our algorithm removes information from high-frequent words, enhancing low-frequent word embeddings. Whitening parameters mainly depend on high-frequent word vectors (Algorithm 1). Thus, this suggests the concern may not materialize empirically; however, extremely low-quality embeddings for rare words might still not improve. Subword-based embeddings could address this. Our response to Reviewer 9YCm under "1. Embedding models" includes results from subword-based fastText embeddings, showing higher performance than non-subword models like GloVe. This indicates the potential value of subword approaches for low-frequent words. We'll incorporate such experiments in our revision, considering the quality aspect you mentioned.
>
> ---
>
> We've attempted to answer within the character limit, but there might be some information gaps. Please feel free to ask for clarification on any points you find unclear.

---

> > ### Comment · Reviewer_jUzu · 2024-08-09
> > **Acknowledgement of Rebuttal**
> >
> > Thank you for your response. I will be maintaining my original assessment and score (but am glad the authors addressed my questions and comments!).

---

### Official Review · Reviewer_fUiC · 2024-07-11

**Soundness:** 2
**Presentation:** 1
**Contribution:** 2
**Rating:** 6
**Confidence:** 2

**Summary:**

Prior work in natural language processing has shown that word embeddings are sometimes concentrated in a small cone of the embedding space.
Prior work has also shown that correcting this can lead to better performance in some downstream tasks.
These prior work, however, do not typically consider a word’s frequency when correcting this issue.
This paper proposes Zipfian whitening: to zero-centre and standardise embeddings while considering their words’ Zipfian frequency.
They show this improves results in a downstream task and then make theoretical arguments for why.

**Strengths:**

The proposed solution is quite simple, and seems to improve results.

The paper provides both empirical support to their proposal, and theoretical arguments in favour of it.

**Weaknesses:**

This paper’s contributions and correctness are hard to assess, in my opinion:
* downstream tasks where experiments are performed are not described in the paper.
* theoretical results are described at a relatively high level (without step-by-step explanations) which make it hard to evaluate its correctness.
* other key information is missing, such as how a single symmetry score is extracted from the symmetry moments to get correlations in Table 2.

Some examples can be found below under “Questions”.

Besides that, as I understand them, experiments are run on a sentence-level similarity tasks. But the proposed methods are at the word level. Why not run word-level evaluation metrics?

Finally, the paper makes the argument that Zipfian whitening makes embeddings’ norms proportional to a word’s information content, while uniform whitening does not. As experiments are performed on sentence-level similarity tasks, this could be an important source of its advantage. Adding experiments with a baseline in which uniform whitening  is performed, but then norms are rescaled based on information content, would be interesting.

**Questions:**

> To evaluate the pre-trained word encoder models and post-processed ones, we used the most commonly utilized downstream tasks in the community, STS-B [7] and SICK-R [23].

1. What are these tasks exactly? It would be helpful if you described them here.
2. Why are you doing sentence-level similarity with word-level embeddings? Why not either: (i) word-level similarity task; (ii) sentence-level embeddings. (The latter should at least be present as a baseline for comparison.)

>  Table 2 lists the correlation coefficients between the symmetry scores and downstream task performance in more detail

How do you compute a single symmetry score? Before (e.g., in Fig 2) you had two separate scores, the first and second moments.

> $∥w∥_{G_w} ≈ 2KL(p(·)∥p(· | w))$


What is $∥w∥_{G_w}$? Did you introduce this already? Also, in the appendix, the proof shows $∥w∥ ≈ 2KL(p(·\mid w’)∥p(· | w))$. Is this a typo, or does the proof work for both cases?

> Another benefit (but slightly more technical) of Zipfian whitening is that we can eventually regard the generative models of a word vector w (given a context c) and a context vector c (given a word w) as being symmetric. This (p(w | c) = p(c | w) can easily be seen from the generative model p(w | c) and the Bayes’ rule, given that the partition function Z z (c) is irrespective of a context c. This symmetry is essential to justify our practice regarding context embeddings being the same as word embeddings.

1. Is this true for Zipfian whitening? Or only for a uniform prior?
2. Why is this beneficial?
3. To the best of my knowledge, word2vec and glove use different embeddings for words and contexts, so this is not exactly true in practice. Do the authors mean that embedding and un-embeddings layers sometimes have shared parameters in language models? That's quite a different setting then what's being analysed here.
4. What does it mean for a context (multi-word) embedding to be the same as a word’s? Maybe making it explicit earlier in this paper that only skipgram- and glove-like models will be analysed (both of whose contexts are assumed to be individual words) would be useful.

**Limitations:**

The authors addressed the paper's limitations.

---

> ### Author Rebuttal · Authors · 2024-08-06
>
> Thank you for your thorough reading and your critical and constructive comments. We especially appreciate your feedback on the clarity and self-contained nature.
>
> ### 1. How to compute symmetry scores in Table 2
>
> We made a typo! Thanks for pointing it out. The labels along the x-axis in Table 2 represent symmetry scores as you may guess. The correct labels for proposed measures are:
> - Incorrect — centering, whitening
> - Correct — 1st moment, 2nd moment
>
> The numbers in Table 2 represent *both* the 1st and 2nd moments of the spatial symmetry, measured with uniform and Zipfian prior. Only when using the Zipfian prior (highlighted in light blue in Table 2), do both the 1st and 2nd moments strongly correlate with the task performance (Line 155–160).
>
> ### 2. The choice of task
>
> > downstream tasks where experiments are performed are not described in the paper.
>
> > experiments are run on a sentence-level similarity tasks. But the proposed methods are at the word level. Why not run word-level evaluation metrics?
>
> This is an important point! Let us address it in detail.
>
> **Datasets we used**: STS-B and SICK-R are both *sentence*-level similarity tasks, which are standard for empirically evaluating the performance of *word* vectors. These datasets consist of pairs of sentences and their semantic similarity rated by annotators. The typical experimental protocol we followed is to sum the word vectors to form a "sentence vector" and then check if the angles between them correlate well with the gold scores.
>
> **Why evaluate word vectors at the sentence-level tasks**: The question "Why not run word-level evaluation metrics?" is a natural and valid inquiry. Our language has a property known as compositionality, which allows infinite semantic content to be conveyed through a finite vocabulary as building blocks. This perspective underlies models like word2vec, BERT, and GPT series, where the fundamental unit of representation is the word; and these models are used to solve tasks with larger components e.g. sentences. Our research adheres to this basic principle of NLP.
>
> **Word-level evaluation**: Your suggestion to evaluate post-processing effects on word vectors using word-level tasks is reasonable! However, existing word-level similarity datasets have significant issues making them less suitable for our work (see Bakarov "A Survey of Word Embeddings Evaluation Methods" 2018, Section 4.1.1). Given that whitening reflects word information content in vector norms, tasks like keyword extraction (which selects words with high information content)  could be good candidates; we'll include such tasks in the camera-ready version.
>
> ### 3. Experiments with a mix of uniform and Zipfian settings
>
> > the paper makes the argument that Zipfian whitening makes embeddings’ norms proportional to a word’s information content, while uniform whitening does not.
> (...)
> Adding experiments with a baseline in which uniform whitening is performed, but then norms are rescaled based on information content, would be interesting.
>
> It's a really interesting idea to isolate the effect of the norm and empirically verify its impact.
> We promptly conducted an experiment similar to Table 1 using GloVe and STS-B.
>
> ||Uniform|Uniform + $\alpha$|Zipfian|
> |:-|:-:|:-:|:-:|
> |+ Centering|41.27|53.66|55.15|
> |+ Whitening|53.22|64.83|70.22|
>
> "Uniform + $\alpha$" refers to the process of "correcting word vectors using a uniform prior, then replacing only the norm with that obtained from Zipfian whitening". We found that appropriate weighting by norm has a critical effect on task performance. It's also interesting that pure Zipfian centering/whitening performs even better. This implies that Zipfian correction has two effects: (i) the *norm* becomes representative of information content (Section 3.1), and (ii) vectors disperse more evenly (isotropic), leading to appropriate positioning w.r.t. *direction* as well. We will incorporate comprehensive results into the manuscript!
>
> ### 4. Notation of norm
>
> > What is $||w||_{G_w}$?
>
> We used $||x||_A$ to denote a norm based on a quadratic form $\sqrt{x^\top Ax}$. We will clarify this point in the manuscript.
>
> > in the appendix, the proof shows $||w|| \approx 2 KL(p(・|w’) || p(・|w))$. Is this a topo, or does the proof work for both cases?
>
> Is Reviewer fUiC referring to Line 435? We also use the notation $||w||_{G_w}$ here rather than $||w||$. If there is a typo elsewhere, we would appreciate it if you could let us know.
>
> ### 5. Symmetry of w and c induced by whitening (Page 7, Footnote 11)
>
> > 1. Is this true for Zipfian whitening? Or only for a uniform prior?
>
> It's true regardless of the prior. You can see this by Bayes' rule: $p(c|w) = p(c)p(w|c)/p(w) = p(c)\exp(\langle w,c\rangle) / Z(w)$, where Eq. (7) and $Z(w)=Z(c)$ is used.
>
> > 2. Why is this beneficial? / 3. To the best of my knowledge, word2vec and glove use different embeddings for words and contexts
>
> Exactly. In standard learning algorithms for static embeddings, asymmetric embeddings are learned although the target co-occurrence distribution is symmetric. What's interesting here is that whitening as a post-processing step can restore this symmetry, which benefits us in approaching the desirable inherent symmetry originally present in the data.
>
> > 4. What does it mean for a context (multi-word) embedding to be the same as a word’s?
>
> In both static and masked/causal models, word prediction is performed through inner products. The "context" can refer to a single word, a cloze sentence, or a prefix of a sentence; abstractly, c encompasses all of these. However, when discussing symmetry as in Footnote 11 for example, it would be clearer to restrict c to co-occurring words.
>
> ---
>
> Thank you once again for your critical reading. Your feedback will help us refine the manuscript to ensure clarity and accuracy. If any remaining points are unclear, please feel free to share your candid feedback.

---

> > ### Comment · Reviewer_fUiC · 2024-08-08
> > **Response to Authors**
> >
> > I thank the authors for their detailed response.
> >
> > I am still not fully convinced by the authors' argument that exclusively evaluating a method proposed for type-level (uncontextual) embeddings on sentence-level tasks is the best choice. The new experiment isolating the effect of the norm on the embeddings' performance is reassuring, though, so I have increased my score.
> >
> > > $∥w∥_{G_w} ≈ 2KL(p(·)∥p(· | w))$
> >
> > Sorry, I should have been more specific here. In the Theorem the KL is between $p(·)$  and $p(· \mid w)$, while the proof in the appendix uses $p(· \mid w')$ as the first term of the KL.

---

> ### Author Response · Authors · 2024-08-13
> **Thank you for your response and clarification!**
>
> ### 2. The choice of task / Word level evaluation
>
> > I am still not fully convinced by the authors' argument that exclusively evaluating a method proposed for type-level (uncontextual) embeddings on sentence-level tasks is the best choice.
>
> Your doubt is entirely justified. Although we're following conventional practices, we're not 100% satisfied with this convention ourselves.
>
> **Lexical similarity**:
>
> Setting aside the criticisms from previous studies for now, we conducted an evaluation using the two most well-known lexical similarity datasets. Below are the correlation coefficients × 100 between the cosine similarity of (corrected) GloVe embeddings and the gold score.
> |WordSim353 (Finkelstein et al. 2002)|Uniform|Zipfian|
> |:-|:-:|:-:|
> |raw|78.70|78.70|
> |+ centering|75.39|79.66|
> |+ whitening|82.31|80.90|
>
> |MEN (Bruni et al. 2012)|Uniform|Zipfian|
> |:-|:-:|:-:|
> |raw|80.49|80.49|
> |+ centering|78.07|80.55|
> |+ whitening|84.35|83.97|
>
> We found that the process of raw $\rightarrow$ Zipfian centering $\rightarrow$ Zipfian whitening consistently improves lexical properties.
>
> Note that, however, the results that "uniform whitening (direction) $>$ Zipfian whitening (direction)" contradicts the experimental results in this rebuttal "3. Experiments with a mix of uniform and Zipfian settings," which showed "**direction: uniform whitening**, norm: Zipfian whitening $<$ **direction: Zipfian whitening**, norm: Zipfian whitening". The cause likely stems from these datasets not being good summaries of natural language, as summarized in Bakarov's "A Survey of Word Embeddings Evaluation Methods" 2018, Section 4.1.1. For instance, the most well-known dataset, WordSim353, consists of only about 200 subjective ratings on common nouns like (tiger, cat, 7.35) or (king, cabbage, 0.23), which may or may not appear in the same document.
>
> **Possible evaluations**:
>
> Through discussions with Reviewer fUiC, two key points became clear: (i) including both word-level and sentence-level evaluations can enhance empirical persuasiveness, and (ii) separating norm and direction allows for more detailed evaluation. The overall picture of feasible evaluation experiments seems to include the following options. We aim to incorporate these aspects as comprehensively as possible in the camera-ready version.
>
> |properties of word embeddings|word-level evaluation|sentence-level evaluation|
> |:-|:-|:-|
> |(whole vector)|analogy|$\checkmark$ STS|
> |norm|keyword extraction|$\checkmark$ STS — isolating the effect of norm|
> |direction|$\checkmark$ lexical similarity|STS — isolating the effect of direction|
>
> ### 6. Complementing the Proof
>
> > In the Theorem the KL is between $p(\cdot)$ and $p(\cdot|w)$, while the proof in the appendix uses $p(・|w’)$  as the first term of the KL
>
> We understand! The proof we included in the Appendix was incomplete. Thank you for pointing this out. We'll clarify below.
>
> First,
> $\lVert \underline{\boldsymbol w} \rVert_{\boldsymbol G(w)}^2$
> at the end of Line 453 is a typo; it should be
> $\lVert \underline{\boldsymbol w' - \boldsymbol w} \rVert_{\boldsymbol G(w)}^2$.
>
> \begin{align}
> 2\mathrm{KL}(p(\cdot\mid w') \\| p(\cdot\mid w))
> &\approx \dots
> \\\\
> &= (\boldsymbol w' - \boldsymbol w)^\top
> \biggl\lbrace\sum_{c \in \mathcal V}p(c\mid w) \boldsymbol c \boldsymbol c^\top\biggr\rbrace
> (\boldsymbol w' - \boldsymbol w)
> \\\\
> &= (\boldsymbol w' - \boldsymbol w)^\top
> \boldsymbol G(w)
> (\boldsymbol w' - \boldsymbol w)
> \\\\
> &= \lVert \underline{\boldsymbol w' - \boldsymbol w}\rVert_{\boldsymbol G(w)}^2
> \text{.}
> \end{align}
>
> The rest, namely
> $2\mathrm{KL}(p(\cdot) \\| p(\cdot\mid w)) \approx \lVert \boldsymbol w\rVert_{\boldsymbol G(w)}^2$,
> follows immediately from the property shown in Appendix K in Oyama et al (*).
> The following are the details.
>
> We can consider a word $w_0$ such that $p(\cdot) = p(\cdot\mid w_0)$, that is, an uninformative word $w_0$ whose presence does not change the marginal distribution at all.
> Next, assuming that the partition function is constant holds up to the first moment, in other words, that Zipfian centering has been performed (**):
>
> $\overline{\boldsymbol w} \coloneqq \sum_{w\in\mathcal V} p(w) \boldsymbol w = \boldsymbol 0$.
> Then,
>
> \begin{align}
> \mathrm{KL}(p(\cdot) \\| p(\cdot\mid w))
> &= \mathrm{KL}(p(\cdot\mid w_0) \\| p(\cdot\mid w))
> \\\\
> &\underset{(\text{Line 453})}{=} \lVert \boldsymbol w_0 - \boldsymbol w\rVert_{\boldsymbol G(w)}^2
> \\\\
> &\underset{(*)}{\approx} \lVert \overline{\boldsymbol w} - \boldsymbol w\rVert_{\boldsymbol G(w)}^2
> \\\\
> &\underset{(**)}{=} \lVert \boldsymbol w\rVert_{\boldsymbol G(w)}^2
> \text{.}
> \end{align}
>
> To be honest, we think we felt "completely done" after writing out the differences from the previous research (Oyama et al.) in the proof. We're glad we were able to update the manuscript.
>
> ---
>
> Once again, thank you for your critical and thorough comments. If there's anything lacking in our response, please let us know.

---

> > ### Author Response · Authors · 2024-08-13
> >
> > (We may have made a mistake with the readers' settings, so we've resubmitted. We are sorry for the increased number of email notifications.)

---

> > ### Comment · Reviewer_fUiC · 2024-08-14
> > **Response to Authors 2**
> >
> > Thanks for this extra set of experiments. I think these extra results are quite interesting (even if partially negative), and agree that incorporating these different evaluation aspects in the camera-ready version would be good. I have increased my score again.

---

### Official Review · Reviewer_VABi · 2024-07-15

**Soundness:** 3
**Presentation:** 4
**Contribution:** 4
**Rating:** 8
**Confidence:** 4

**Summary:**

This paper proposes "Zipfian whitening" of word vectors, that is, taking
word probability in consideration when taking averages for whitening them.
In addition to presenting the proposed simple algorithm and experimentally
evaluating in suitable NLP tasks, this paper also introduces measure of
isotropy of word vectors, and theoretically explains why uniform averaging
does not work well in practice.

**Strengths:**

Basically this is a good paper, and matches well with NeurIPS because the
proposed whitening and the geometric discussion also applies to other fields
than natural language processing.
I would like the authors to include some actual words uniformly sampled from
the vocabulary, which clearly shows averaging uniformly with such (mostly rare)
words is a bad idea.

**Weaknesses:**

My only concern is the title and assumption: why "Zipf"? Zipfian distribution
means that the probability of each item decays inversely proportional with
the rank, yielding a heavy-tailed distribution. However, this kind of Zipfian
characteristics does not seem to be used in the theory: it is just a
"Expected whitening" rather than "Zipfian whitening".

Actually many distributions, including words, have Zipfian property, thus it
is interesting to see, empirically and/or theoretically, if the proposed method
works for non-uniform, but non-Zipfian distributions.
Without such considerations, the title of "Zipfian whitening" might be
misleading as a scientific research.

Minor
- Figure 2: performance of each configuration could be displayed as the size of
each disk. This does not need color printing, and humans generally have more
senses over the difference of sizes over difference of intensities.
- p9: frequency We -> frequency. We

**Questions:**

Nothing.

**Limitations:**

No problems.

---

> ### Author Rebuttal · Authors · 2024-08-06
>
> We're pleased to receive your positive evaluation! We intend to address all the points you've raised.
>
> ### 1. Qualitative demo of the unnaturalness of uniform word distribution
>
> > I would like the authors to include some actual words uniformly sampled from the vocabulary, which clearly shows averaging uniformly with such (mostly rare) words is a bad idea.
>
> Thank you for this excellent suggestion! Given that NLP is inherently driven by real data, incorporating concrete examples as a qualitative evaluation can likely provide readers with a more intuitive understanding of our paper's idea. We've quickly performed a sampling using the word frequencies employed in our paper:
>
> - uniform sampling: `['scintillation', 'fanon', 'rubato', 'upstanding', 'collard', 'creeks', 'skookum', 'unbelievers', 'monocyte', 'nishikawa', 'crusher', 'gerwen', 'abrah', 'silverchair', 'hangman', 'unitary', 'klausen', 'arousal', 'heat', 'bridgnorth', 'mildred', 'porton', 'aquasox', 'wylie', 'hipaa', 'krimuk', 'hexahedron', 'kuei', 'barbera', 'dalvi', 'gilding', 'visakhapatnam', 'tatsuo', 'tarascon', 'bajram', 'scholes', 'hadad', 'incidental', 'theodosius', 'reichskommissariat', 'boeheim', 'amsl', 'buencamino', 'thrasyvoulos', 'insulated', 'discourtesy', 'nisra', 'ycko', 'luen', 'dooku']`
> - Zipfian (frequency-aware) sampling: `['nine', 'ranked', 'zero', 'the', 'garcia', 'rank', 'station', 'the', 'for', 'four', 'williams', 'drunken', 'a', 'one', 'eight', 'of', 'were', 'zero', 'debate', 'orchestra', 'of', 'wrist', 'points', 'fractured', 'the', 'to', 'redirect', 'adnan', 'white', 'car', 'fond', 'concluded', 'under', 'two', 'by', 'five', 'his', 'infection', 'the', 'the', 'pop', 'in', 'one', 'in', 'one', 'one', 'fram', 'handled', 'battle', 'mutual']`
>
> The latter clearly seems to capture a more "natural" representation of language as we typically encounter it in text, while the former uniform sampling is likely to give an impression quite detached from human language. We will include this comparison in the section where we propose calculating expectations weighted by frequency.
>
> ### 2. The naming of "Zipfian"
>
> > My only concern is the title and assumption: why "Zipf"? Zipfian distribution means that the probability of each item decays inversely proportional with the rank, yielding a heavy-tailed distribution. However, this kind of Zipfian characteristics does not seem to be used in the theory
>
> Your point is well taken. Thank you for bringing this to our attention. Our focus is on the mismatch between word frequency distribution and uniform distribution, and we haven't developed an argument dependent on the degree of tail heaviness. We used the well-known example of "Zipf" because power-law distributions are common in the real world, and we're specifically dealing with word frequencies. You're right that it would be better to make the method name scientifically accurate. We appreciate your suggestion of "expected whitening," and alternatives like "frequency-aware whitening" or "distribution-aware whitening" could also be considered. In any case, we will aim for a name that truly reflects the essence of the method.
>
> ---
>
> Thank you also for your comments on the presentation and typo! We'll address these as well.

---

> > ### Comment · Reviewer_VABi · 2024-08-10
> > **Reply to "Zipfian"**
> >
> > As the title of the paper, "Zipfian whitening" is appealing, while "Expected whitening" is clearly dull.
> > Therefore, I think it might suffice to include some explanation that "Zipfian" is a kind of jargon that actually represents
> > non-uniform word distribution. Besides, I would also like to know what would occur if the distribution is non-uniform but
> > not so much Zipfian.

---

> ### Author Response · Authors · 2024-08-13
> **Thank you for your additional comments!**
>
> ### 2. Naming of Zipfian
>
> We were also fond of the name "Zipfian whitening," so the direction you've suggested is ideal.
>
> \# we often hear "long-tailed" as a term representing non-uniformity as well.
>
> In any case, we will make sure to add a note to ensure scientific accuracy.
>
> ### 3. Experiments with non-uniform and non-Zipfian data
>
> > Besides, I would also like to know what would occur if the distribution is non-uniform but not so much Zipfian.
>
> Indeed, experiments with such data could lead to an interesting message like "the further the empirical distribution deviates from uniform, the more standard centering/whitening suffers from distribution mismatch."
>
> However, at least for natural language data, there's a universal tendency for various phenomena to follow power-law distributions (Zipf's law, Heaps' law, etc.), making it challenging to find situations that don't adhere to power-law distributions.
> As an alternative, we could consider examples of representation learning models like word2vec, BERT, and causal LMs being utilized in other domains. For instance, item2vec in recommender systems (Barkan and Koenigstein, RecSys 2016), metapath2vec in heterogeneous information networks (Dong et al. KDD 2017), and recent Transformer models for time series might naturally have frequency distributions that aren't heavy-tailed. We plan to elaborate on this at least in the future work section.
>
> Another option could be to experimentally adjust the heaviness of the frequency distribution used for calculating expectations (although this would be a pseudo-verification since the frequency distribution of the corpus used for representation learning would still follow a power law). For example, we could vary $n$ in $p_{\text{new}}(w) \propto p(w)^n$. We intend to conduct experiments in this direction. Regarding the choice of frequency distribution, we also received comments from 9YCm and conducted a brief verification in the "2. Word frequency" section of the rebuttal, for your reference.

---

### Decision · Program_Chairs · 2024-09-25

**Decision:**

Accept (poster)

**Comment:**

The paper proposes using the information of word frequency in in post-processing word embeddings, and provides both theoretical and empirical justifications of its benefits. The reviewers liked the simplicity and the theory components of the work, though some remain a bit unconvinced of certain parts of the paper and its experimental settings (still leaning positive). My only real concern is its relatively limited scope of correcting the word embedding space, but the paper seems to make sufficiently strong contributions in novelty and rigor in this scope to justify an accept.